# Towards understanding multimodal in-context learning

## Abstract

Multimodal large language models (MLLMs) often exhibit in-context learning (ICL) abilities, yet the conditions under which multimodal ICL emerges, and the mechanisms underlying it, remain poorly understood. In particular, how training data statistics and architectural choices jointly shape this capability is still an open question. To address this, we reverse-engineer multimodal ICL by training small transformer models on controlled synthetic classification tasks with varying data statistics and architectural choices. We begin by revisiting core principles of unimodal ICL in *modern* transformers. While several prior findings replicate, our experiments yield two notable observations. First, Rotary Position Embeddings (RoPE), a standard component in contemporary LLMs, can delay the onset of ICL circuits. Second, larger models require stronger statistical cues in the training data for strong ICL to appear. Extending our analysis to the multimodal setting reveals a fundamental learning asymmetry. Once a primary modality has learned a core ICL circuit from statistically diverse data, a secondary modality can reach comparable ICL performance with far less data complexity. In contrast to the unimodal regime, we further find that model scaling consistently improves multimodal ICL. To understand why these patterns emerge, we turn to mechanistic analysis. Using progress measures that track circuit formation during training, we show that ICL accuracy is tightly correlated with the strength of an induction-style circuit that copies labels from in-context exemplars that match the query. Both unimodal and multimodal ICL rely on this induction mechanism, while multimodal training primarily refines and extends it across modalities. Together, these results provide a mechanism-level account of ICL in modern multimodal transformers, offer explanations for several empirical phenomena observed in MLLMs, and introduce a controlled testbed for future work on multimodal ICL.

## 1 Introduction

In Large Language Models (LLMs), in-context learning (ICL) is the ability to perform new tasks by learning from input-output demonstrations provided in the context, without parameter updates (Brown et al., 2020). MLLMs extend this to interleaved image-text demonstrations, enabling cross-modal reasoning from contextual examples alone (Alayrac et al., 2022; Abdin et al., 2024).

While unimodal ICL has been widely studied, including mechanistic analyses that tie ICL to properties of the training distribution (Chan et al., 2022; 2024) and to specialized *induction circuits* that retrieve relevant context examples and copy their labels (Olsson et al., 2022; Reddy, 2024), two critical gaps remain. First, most mechanistic insights come from simple attention-only transformers that lack many architectural components of modern LLMs, such as RoPE and layer normalization. Second, the extension from unimodal to multimodal settings introduces fundamental questions about how ICL circuits develop when models must integrate information across different modalities with different statistical properties.

Meanwhile, in the multimodal ICL domain, existing work has largely prioritized empirical evaluation (Zhao et al., 2022; Zong et al., 2024) or high-level analyses of retrieval strategies and prompt design (Qin et al., 2024; Xu et al., 2024). Although recent studies highlight that MLLMs often neglect the visual modality (Baldassini et al., 2024; Chen et al., 2025b), these works treat the model as a black box. The circuit-level implementation of multimodal ICL remains unexplored.

Figure 1: **(a)** The context consists of $N$ triplets followed by the target query. The paired examples $(x_i, x_i')$ from two modalities, with a shared label $l_i$, are generated from Gaussian Mixture Models (GMMs) by controlling within-class variation $\varepsilon_1$ and $\varepsilon_2$. **(b)** The distributional properties for the synthetic data. The burstiness $B$ determines repeated class occurrences in context. Class frequencies follow a Zipfian distribution with exponent $\alpha_1$ and $\alpha_2$. **(c)** Evaluation distinguishes between IWL, where target queries belong to class seen during training while not in the context during evaluation, and ICL, where target queries are novel but in the context. A swapped-label condition further isolates ICL by permuting the labels.

This paper closes these gaps by providing a systematic, reverse-engineering account of ICL in modern, multimodal transformers. Our investigation addresses three central questions: *(1) Do the statistical principles governing ICL in simplified transformers transfer to modern LLM-style architectures? (2) How do these effects carry over from unimodal to multimodal settings? (3) What are the underlying attention circuits that implement multimodal ICL, and how do they relate to their unimodal counterparts?* To answer these questions, we train small but architecturally modern transformer models on controllable synthetic classification tasks. This controlled environment allows us to systematically vary data statistics and architectural components in both unimodal and multimodal regimes. We employ interpretable progress measures that track the formation of key attention patterns related to ICL throughout training, enabling us to quantitatively link model behavior to circuit-level dynamics.

We report the following key findings: (1) **Modern architectures can impair unimodal ICL.** While the statistical drivers of ICL transfer to modern architectures, RoPE can delay the formation of induction heads. Furthermore, scaling up unimodal models can paradoxically favor memorization over in-context adaptation given fixed training data complexity(Sec. 3). (2) **Multimodal ICL exhibits a strong learning asymmetry.** A primary modality trained on high-diversity data learns the core ICL circuit, which a secondary modality can then leverage with surprisingly low data complexity (Sec. 4.1). In direct contrast to the unimodal case, scaling consistently improves multimodal ICL (Sec. 4.2). (3) **Secondary-modality representation quality is the bottleneck.** Multimodal ICL performance is limited primarily by the quality of the secondary modality's representations and their alignment to the decoder's embedding space. A pretrained encoder for this modality is essential (Sec. 4.3). (4) **Induction circuit strength tracks ICL accuracy, and multimodal training refines existing circuits.** Using progress measures, we show that induction circuit strength is tightly correlated with ICL accuracy in both settings. The same induction circuits from unimodal learning underlie multimodal ICL; multimodal training primarily refines the induction head responsible for label matching (Sec. 4.4). (5) **Validation on real data and MLLMs**. We validate our distributional findings and encoder-quality predictions on Omniglot (Lake et al., 2019) and our mechanistic findings on MLLMs, including Qwen2.5-VL. Our mechanistic account also generates hypotheses that explain previously puzzling phenomena in real MLLMs (Sec. 4.5).

## 2 PRELIMINARIES

Understanding how multimodal ICL emerges requires isolating the responsible mechanisms. Studying a full MLLM stack entangles many moving parts, making causal attribution difficult. We therefore adopt a controlled synthetic approach that enables precise manipulation of data statistics and architectural components while maintaining mechanistic interpretability.

**Task description.** Let $\mathcal{X}_1$ and $\mathcal{X}_2$ denote the input spaces of two modalities and let $\mathcal{L} = \{L_1, \ldots, L_n\}$ be the shared label set. We feed the model with a context consisting of $N$ labelled

exemplars followed by an unlabelled query. In the unimodal setting, we follow Reddy (2024), with context comprising $N$ item–label pairs from a single modality: $x_1, \ell_1, x_2, \ell_2, \ldots, x_N, \ell_N, x_q$. Each $x_i \in \mathcal{X}_1$ is an example whose ground-truth label is $\ell_i \in \mathcal{L}$. The model must predict $\ell_q$ for the query item $x_q$. In the multimodal setting, we extend the task by presenting paired exemplars from two modalities: $x_1, x_1', \ell_1, x_2, x_2', \ell_2, \ldots, x_N, x_N', \ell_N, x_q, x_q'$ (Fig. 1a). Here $x_i \in \mathcal{X}_1$ and $x_i' \in \mathcal{X}_2$ correspond to the same label $\ell_i$. Importantly, at least one exemplar (unimodal) or exemplar pair (multimodal) in the context shares the query's class, ensuring that ICL is in principle possible.

**Synthetic data generation.** We consider two modalities and generate data for both modalities from Gaussian Mixture Models (GMMs), allowing for precise control over data properties. For modality $m \in \{1, 2\}$, we sample from a GMM with $K_m$ classes in $\mathbb{R}^{D_m}$. Class prototypes are $\mu_k \sim \mathcal{N}(0, I_{D_m}/D_m)$, and class instances are generated by:

$$x_i = \frac{\mu_k + \varepsilon_m \eta}{\sqrt{1 + \varepsilon_m^2}}, \quad \text{where} \quad \eta \sim \mathcal{N}(0, I_{D_m}/D_m). \tag{1}$$

The parameter $\varepsilon_m$ sets the within-class variability (Fig. 1a), with rescaling factor ensuring that $\|x\| \approx 1$. Each modality has $K_m$ classes mapped many-to-one into the shared label set $\mathcal{L}$. This models fine-grained subclasses collapsing to a common output vocabulary. Each label is also associated with a prototype vector sampled from $\mathcal{N} \sim (0, I_D/D)$. In the multimodal setting, the two generators are dependent. Specifically, we set the label space $\mathcal{L}_2 \subset \mathcal{L}_1$ to reflect common MLLM practice where the decoder generates in the primary modality's vocabulary (e.g., text), so the secondary modality aligns to that vocabulary rather than expanding it.

**Parameterizing the data distribution.** Beyond the intrinsic GMM parameters $(D, K, \varepsilon)$, we also control sequence-level statistics to study ICL dynamics (see Fig. 1b). The *burstiness $B$* (Chan et al., 2022) creates contexts with $N/B$ classes, each appearing $B$ times. This models the multi-shot demonstrations common in real prompts. $p_B$ determines how often such bursty sequences occur: with probability $p_B$ an input is generated in the bursty manner as described above, whereas with probability $1 - p_B$ the context items and the query are sampled i.i.d. across classes. Finally, *Zipfian skew $\alpha$* (Piantadosi, 2014) parameterizes long-tailed class frequencies $f(k) \sim k^{-\alpha}$ typical of web corpora and image datasets, determining the balance between frequent and rare concepts.

**Evaluation metrics.** We explicitly separate two learning mechanisms to avoid misattributing memorization to contextual reasoning, similar to Reddy (2024): In-Weight Learning (IWL) assesses performance on test sequences sampled i.i.d. from the training distribution, measuring knowledge stored in parameters while ICL assesses performance on sequences with novel classes, forcing reliance on contextual examples rather than memorized associations. As an additional ICL metric, we evaluate the model on sequences in which the context labels are swapped relative to training, invalidating the memorized mapping. We illustrate the evaluation in Fig. 1c and App. A.1.2.

# 3 Establishing Architectural Premises: ICL in Modern Transformers

While previous work (Reddy, 2024) has revealed foundational properties of ICL in attention-only transformers, such as dependency on specific data distribution, it remains unclear how these principles operate within modern, LLM-style configurations. Here, we construct a controlled setting to isolate the data-centric and architecture-centric factors that drive ICL, providing premises for our later multimodal study.

Concretely, we move beyond the canonical 2-layer attention-only setup and adopt a two-layer transformer decoder architecture incorporating RMSNorm (Zhang & Sennrich, 2019), SiLU activation (Elfwing et al., 2018), and optionally RoPE (Su et al., 2022), which are components characteristic of contemporary LLMs (Touvron et al., 2023). We systematically vary the statistical properties of the training data and model size, enabling clean attribution of how each factor shapes the emergence of ICL.

## 3.1 Data statistical principles hold within modern architectures.

All key findings revealed in (Reddy, 2024; Chan et al., 2022) are reproduced: increasing the number of classes $K$, burstiness $B$, or within-class variation $\varepsilon$ promotes ICL, while increasing the Zipf

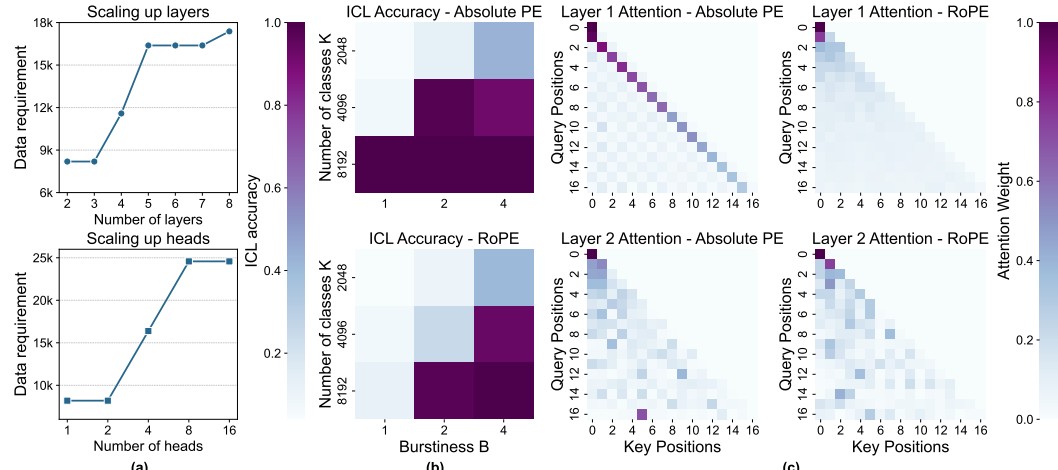

Figure 2: **In unimodal transformers, scaling up and RoPE suppress ICL.** (a) The data complexity (proxy $K \cdot \sqrt{B}$) required for model of different sizes to achieve the perfect ICL accuracy ($> 0.95$). (b) Across data regimes, RoPE yields lower ICL accuracy than absolute positional encodings (PE). (c) Attention maps for an example with the correct label at position 5: absolute PE shows clear previous-token and induction heads; with RoPE these patterns are diminished.

exponent $\alpha$ shifts performance toward IWL, with an optimal balance achieved when $\alpha = 1$ (see App. A.2.1). We also recover the ICL–IWL trade-off: settings that strengthen ICL correspondingly diminish IWL, and vice versa. Together, these findings confirm that the core data statistical drivers are architecture-agnostic.

### 3.2 MODEL SCALING RAISES THE ICL THRESHOLD.

We systematically scale model depth or attention heads and train the model until convergence, using the data with the same statistics cues. We reveal a fundamental tension between model capacity and ICL utilization. While both small and large models can learn ICL (App. A.2.1), larger models require substantially stronger statistical cues in training data to achieve the same ICL accuracy (Fig. 2 a). This requirement grows faster with the number of heads than with the number of layers. We attribute this to how multi-head attention distributes capacity: more heads allow information to be partitioned into specialized subspaces, enabling the item–label memorization. This creates a low-loss shortcut that competes with the ICL solution. Adding layers increases depth but does not create as many independent storage slots for memorization, so the bias toward in-weight learning is weaker. This interpretation aligns with prior observations that induction-style behavior typically emerges in a subset of heads while other heads specialize in memorization (Elhage et al., 2021; Singh et al., 2024). Importantly, *scaling does not eliminate ICL capacity*; it raises the threshold of statistics cues needed for the ICL solution to outcompete memorization.

### 3.3 RoPE DELAYS THE ONSET OF ICL.

Switching from absolute positional encodings (APE) to RoPE causes a marked drop in ICL accuracy (Fig. 2b). Although RoPE is widely adopted for its strong length generalization, in our setting this benefit coincides with a consistent degradation of multimodal ICL. Attention visualizations show that the model struggles to form strong previous-token heads with RoPE, and the induction head is also less clear (Fig. 2c), which we will investigate in Sec. 4.4. We hypothesize this is because RoPE's multiplicative rotational structure lacks the discrete, offset-based cues of absolute encodings, making it harder for the model to learn the simple token-copying operations crucial for ICL. For completeness, we also evaluate ALiBi (Press et al., 2021) and Hybrid PE. We provide detailed results in App. A.2.1, which show that the design of RoPE and ALiBi provides a weaker inductive bias for the simple, offset-based induction circuits. This effect is most pronounced in low-data-complexity regimes, and *sufficiently high data complexity can compensate for this weaker bias. Importantly, they do not prevent ICL.*

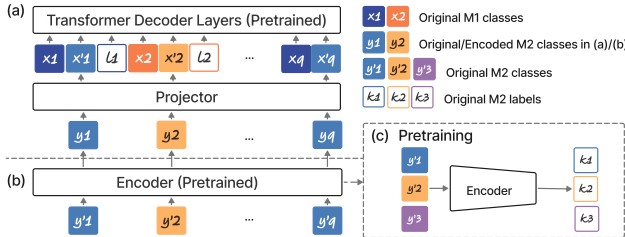
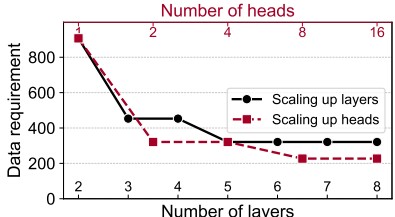

Figure 3: Multimodal setup. M1 denotes the base modality and M2 the additional modality. *(a) Projector-only setup*: an MLP projector aligns M2 features to the M1 embedding space. *(b) Encoder-augmented setup*: a pretrained M2 encoder is stacked before the projector and decoder. *(c) Encoder pretraining*: the M2 encoder is pretrained on M2-specific classes/labels.

Figure 4: Contrary to the unimodal finding, larger decoders achieve the same ICL accuracy with lower data requirements in multimodal models (measured by $K_2 \cdot \sqrt{B}$).

> **Takeaway**: Data-property effects on ICL persist in modern LLM-style transformers. Scaling shifts learning toward in-weight memorization and raises the data complexity needed for model to choose ICL. RoPE further suppresses ICL by disrupting induction circuits.

## 4 MULTIMODAL IN CONTEXT LEARNING

Building on our unimodal ICL baseline, we now investigate the multimodal setting to understand how ICL principles transfer across modalities. We employ a two-stage training process (Fig. 3). First, we pretrain a transformer decoder, same as the unimodal setting, on a single primary modality (M1). Next, we introduce a secondary modality (M2) by adding a simple MLP projector to map M2 features into M1's embedding space. We then jointly train it with the projector. As an optional extension, we insert a pretrained M2 encoder before the projector to probe representation quality. Our analysis continues to focus on data statistics and architectural choices.

### 4.1 DATA ASYMMETRIES REVEAL A PRIMARY AND SECONDARY ROLE FOR MODALITIES.

We examine how the data distributions of two modalities affect ICL. Firstly, the results (Fig. 5a) show that increasing the class number, burstness, and diversity of M2 increases ICL, similarly to the unimodal setting. Moreover, our experiments reveal a fundamental asymmetry: after pretraining the decoder on a high-diversity M1 ($K_1 = 8192$), M2 requires surprisingly little class diversity to achieve comparable ICL; a relatively small $K_2 = 256$ is sufficient (Fig. 5a). This suggests that M1's role is to install the core ICL circuit, while M2's primary role is simply to provide a distinguishable signal that the projector can map onto the decoder's pre-existing feature space. Further experiments support this view. While within-class variation $\varepsilon$ in both modalities boosts ICL, increasing it for M2 has a much stronger positive effect (Fig. 5b). Since M1 is already well-represented from pretraining, higher $\varepsilon_2$ is more critical as it forces the model to learn a robust and generalizable mapping for the new modality. Finally, performance is optimized when class-frequency skews ($\alpha$) match across modalities: pretraining M1 with $\alpha_1 \approx 1$, the Zipfian skew characteristic of natural-language corpora used to pretrain decoders (Piantadosi, 2014), yields the best multimodal balance when M2 also has $\alpha_2 \approx 1$, consistent with the long-tailed concept distributions in large image and image–text datasets (Changpinyo et al., 2021; Thomee et al., 2016; Udandarao et al.) (Fig. 5c). This parallels standard MLLM practice: a text-pretrained decoder is adapted with long-tailed vision(–language) data, and aligning skews reduces cross-modal mismatch and improves transfer.

> **Takeaway.** With a decoder initialized from unimodal pretraining, strong multimodal ICL is achievable without highly complex data from the second modality: moderate class diversity, moderate burstiness, and sufficient within-class variability in the second modality suffice.

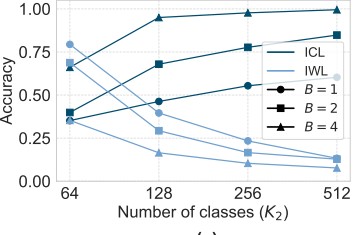 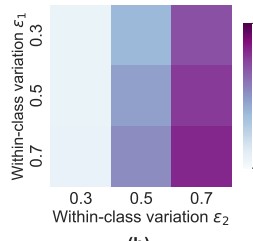 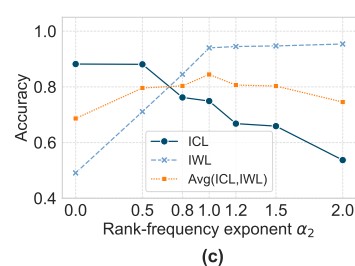

(a)  (b)  (c)

Figure 5: Multimodal data asymmetry: when the decoder is pretrained on a diverse primary modality, only moderate class diversity/burstiness in the secondary modality is needed, and aligning long-tail skews further stabilizes transfer. (a) With $K_1$=8192, ICL increases with $K_2$ and burstiness $B$; even $K_2$=256, $B$=4 attains near-perfect ICL. (b) Raising $\varepsilon_2$ benefits ICL markedly more than raising $\varepsilon_1$. (c) Fixing $\alpha_1$=1, the ICL–IWL balance is best when $\alpha_2 \approx 1$.

## 4.2 THE DECODER'S ARCHITECTURE HAS CONTRASTING EFFECTS ON MULTIMODAL ICL

**Scaling up improves multimodal ICL.** In contrast to our unimodal finding, scaling up the decoder consistently improves multimodal ICL, allowing ICL to emerge with less complex data cues (Fig. 4). Since the decoder already learns ICL over M1, scaling primarily increases representational bandwidth for integrating M2 via the projector, rather than relearning ICL from scratch. This allows larger models to generalize more effectively from the multimodal training data.

**RoPE continues to delay the onset of multimodal ICL.** While our unimodal findings showed that ICL relies on a circuit of previous-token and induction heads, a key question is whether these same mechanisms are responsible for multimodal ICL. As shown in Fig. 6, models with absolute PE successfully form the characteristic previous-token and induction head patterns, confirming that the core ICL mechanism transfers directly. In contrast, and consistent with our unimodal results, RoPE significantly damages this circuit (Fig. 6). The potential underlying cause remains the same: RoPE's relative encoding interferes with the absolute offset-based cues essential for induction heads to function correctly. Similarly, with an increase in data complexity, strong ICL can still be achieved. We will discuss more on the induction circuit in detail in Sec. 4.4.

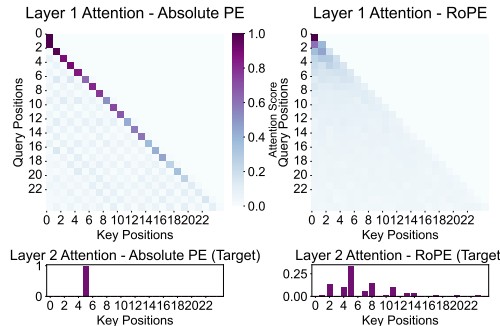

Figure 6: In a multimodal example with the correct label at position 5, absolute positional encodings yield clear previous-token and induction heads, whereas **RoPE blurs these induction patterns.**

> **Takeaway.** Scaling up the decoder size in the multimodal model improves ICL. Same as the unimodal case, RoPE increases the threshold for strong multimodal ICL.

## 4.3 THE ROLE OF THE ENCODER AND ITS IMPACT ON FEATURE REPRESENTATIONS

While attention mechanisms drive ICL, they can only operate effectively on the representations they receive. This raises a fundamental question: how does the quality of input representations affect multimodal ICL performance?

**High-Dimensional M2 features hurt ICL.** Our initial experiments revealed a consistent drop in ICL accuracy as M2 feature dimensionality increased without any dedicated processing (Figure 7a). This suggests that models struggle to extract task-relevant structure from high-dimensional inputs when relying solely on the MLP projector.

**Pretrained encoders dramatically improve ICL across all training regimes.** To address the above limitation, we introduce an encoder pretrained on M2 data (Fig. 3c). We connect this encoder to the pre-trained decoder via an MLP projector (Fig. 3b). We experiment with three training strate-

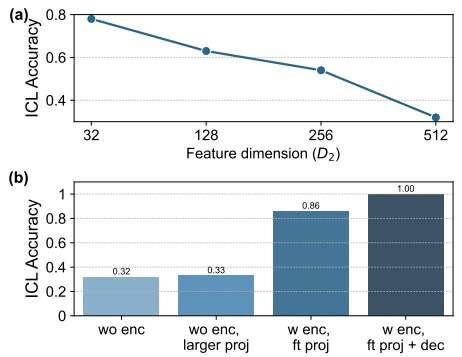
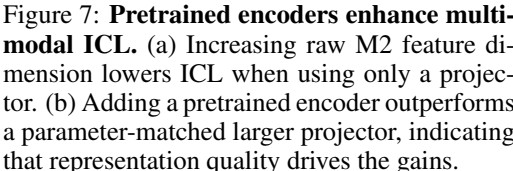
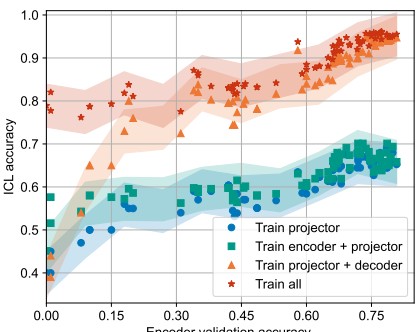

Figure 7: **Pretrained encoders enhance multimodal ICL.** (a) Increasing raw M2 feature dimension lowers ICL when using only a projector. (b) Adding a pretrained encoder outperforms a parameter-matched larger projector, indicating that representation quality drives the gains.

Figure 8: **Encoder quality predicts downstream ICL.** Multimodal ICL on Omniglot increases with the encoder's validation accuracy; gains saturate only when the decoder is frozen, while joint training of the decoder continues to yield improvements.

gies: (1) training only the projector, (2) training the projector & pretrained decoder, and (3) training all the components. Results in Fig. 7b indicate that introducing a pretrained encoder significantly improves ICL accuracy across all training regimes. Importantly, this improvement is not simply due to increased parameter count—when we scale up the MLP projector to match the total parameters of the encoder-equipped model, the pretrained encoder still consistently outperforms the enlarged projector (Fig. 7b). This confirms that encoders contribute through improved representation quality and cross-modal alignment, not just additional capacity.

**Encoders reshape the feature space.** To probe how the encoder facilitates ICL, we quantified its effect on the geometry of M2 class prototypes. The encoder compresses the M2 features into a compact and structured subspace. This is accompanied by better cross-modal alignment with M1 features: centered kernel alignment between M2 and M1 increases from 0.12 to 0.14, and the $L2$ distance decreases from 1.47 to 1.37 when an encoder is employed. These changes indicate that encoders produce representations more compatible with the pretrained decoder's latent space.

**Validation on real data reveals the importance of M2 representation quality.** While synthetic data offers fine-grained control over distributional properties, it is significantly easier to classify compared to real-world inputs. To further investigate the role of the encoder under more realistic conditions, we extend experiments to Omniglot (Lake et al., 2019) images as M2. We first show that all the data distributional findings in Sec. 3 transfer to real images (App. A.3.1), and then train encoders with varying difficulty levels (detailed in App. A.3.2) and measure their impact on downstream multimodal ICL. As shown in Fig. 8, encoder validation accuracy strongly predicts multimodal ICL performance, confirming that perception quality is the key bottleneck, which aligns with recent empirical observations by Zong et al. (2024). Moreover, joint fine-tuning of the encoder, projector, and decoder consistently yields the best performance, indicating the necessity of updating all components to unlock full downstream ICL performance.

> **Takeaway.** A pretrained encoder compresses high-dimensional M2 inputs to better align with the decoder's latent space. Encoder quality strongly predicts downstream ICL capability, confirming that perception is a key bottleneck. Critically, joint optimization of all components is necessary to fully unlock the top performance.

## 4.4 QUANTIFYING ICL MECHANISMS THROUGH PROGRESS MEASUREMENTS

### 4.4.1 PROGRESS MEASUREMENTS

Prior work on simple transformers identified a two-step ICL circuit: an early-layer "previous-token head" copies information, and a later-layer "induction head" uses it to find matching examples and retrieve their labels (Olsson et al., 2022). However, these circuits can be difficult to detect when attention patterns are diffuse, particularly with relative position encodings like RoPE (Sec. 3.2). To

quantify the circuit-level dynamics of ICL even when the patterns are not clearly visible, we employ a suite of "progress measurements" designed to track the formation of key attention patterns, inspired by (Reddy, 2024). The attention at layer $m$ is denoted by $\text{Attn}_m$.

**Previous Token Head Strength** (PHStrength) measures attention paid by all the tokens to their immediate predecessor. Given the sequence length $L_{seq}$, we define:

$$\text{PHStrength}_m^{(1)} = \frac{1}{L_{\text{seq}} - 1} \sum_{i=1}^{L_{seq}-1} \text{Attn}_m\big(\text{query}_i \to \text{key}_{i-1}\big). \tag{2}$$

For multimodal settings, we also define $\text{PHStrength}_m^{(2)}$ for attention to tokens two positions earlier, allowing us to detect shifts in attention span.

**Induction Head Strength** (IndStrength) quantifies how strongly the target token attends to the labels of the context examples of the same class.

$$\text{IndStrength}_m = \frac{1}{|\mathcal{P}|} \sum_{j \in \mathcal{P}} \text{Attn}_m(\text{query}_{\text{target}} \to \text{key}_j), \tag{3}$$

where $\mathcal{P}$ is the set of positions immediately after context examples of the same class as the target.

**Target Label Association** (TLA) measures total attention the target token paid to all context label positions.

$$\text{TLA}_m = \sum_{j \in \mathcal{Y}} \text{Attn}_m\left(\text{query}_{\text{target}} \to \text{key}_j\right), \tag{4}$$

where $\mathcal{Y}$ is the set of positions of all the labels in the context.

**Context-label accuracy** (CLA) measures if the predicted label appeared in the context, indicating reliance on context. Let $\hat{y}$ be the predicted label, $\{y_i\}_{i=0}^N$ the labels of the $N$ context examples, then

$$\text{CLA} = \mathbb{P}\left(\hat{y} \in \{y_i\}_{i=1}^N\right). \tag{5}$$

Except for Context Label Accuracy, all metrics are computed separately for each transformer layer.

### 4.4.2 ICL DYNAMICS: FROM FORMATION TO REFINEMENT

We analyze the relationship between our progress measurements and ICL accuracy across both training stages. The results reveal that ICL accuracy is highly correlated with thw strength The results reveal a clear narrative: unimodal pretraining is responsible for forming the foundational ICL circuit, while subsequent multimodal training primarily refines the label-matching component of this existing circuit.

**Correlation analysis reveals a shift in ICL's primary driver.** We compute the Pearson correlation between metrics and ICL accuracy. The results in Tab. 1 differ significantly between the two settings. In the unimodal pretraining, $\text{PHStrength}_1^{(1)}$ (previous-token copying) and $\text{IndStrength}_2$ (label matching) show the strongest correlation with ICL accuracy. $CLA$ (predicting the label from the context) is also a strong correlate. This confirms that initial ICL development is driven by learning the foundational mechanisms of an induction circuit: learning the "looking back" and "copy" operation, identifying that labels come from the context, and learning the "match" operation.

However, in the multimodal setting, the dynamics shift. We find that $\text{IndStrength}_2$ becomes the strongest correlate of ICL accuracy. Conversely, the correlation for $CLA$ drops to just 0.02. As shown in Fig. 9, this is because the model has already mastered this behavior and $CLA$ is consistently high from the start of multimodal training, so it is no longer a driver of further improvement. $\text{PHStrength}_1^{(1)}$ remains a strong correlate, indicating the base "copying" mechanism is stably reused. This shift suggests that the model reuses the foundational behaviors learned in pretraining and the new bottleneck for performance is applying this circuit to M2. The model must learn to match the M2 query feature to the context examples, and this "matching" skill is precisely what $\text{IndStrength}_2$ quantifies. This is also evidenced by the consistently low correlation for $\text{PHStrength}_1^{(2)}$, which indicates that the model *does not* ignore M2 data by only examining M1 and labels as the unimodal stage. In App. A.3.3 we provide a more detailed analysis.

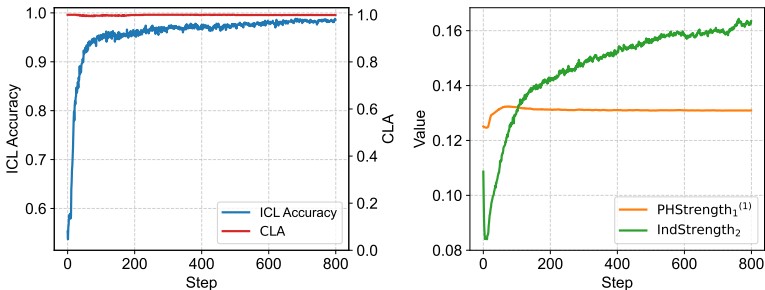

Figure 9: Comparison of the previous token head and induction head with ICL accuracy in the multimodal setting. The previous token head is learned from the unimodal stage and slightly refined while the refinement of the induction head is the main driver. CLA stays at a very high value since the beginning of the multimodal training after being leaned from unimodal stage.

Table 1: Pearson correlations ($\rho$) between top progress measurements ($\rho \geq 0.5$) and ICL accuracy. In the unimodal setting, learning foundational mechanisms like previous-token copying and context reliance is dominant. In the multimodal setting, the focus shifts to refining the label-matching induction head. The full table is in the App. A.3.3.

<table>
<tr><td colspan="3">(a) Unimodal Setting</td></tr>
</table>

(a) Unimodal Setting

| Rank | Metric | $\rho$ |
|------|--------|--------|
| 1 | $\text{PHStrength}_1^{(1)}$ | 0.72 |
| 2 | CLA | 0.65 |
| 3 | $\text{IndStrength}_2$ | 0.61 |
| 4 | $\text{TLA}_1$ | 0.59 |

(b) Multimodal Setting

| Rank | Metric | $\rho$ |
|------|--------|--------|
| 1 | $\text{IndStrength}_2$ | 0.70 |
| 2 | $\text{PHStrength}_1^{(1)}$ | 0.58 |
| 3 | $\text{TLA}_2$ | 0.56 |
| 4 | $\text{TLA}_1$ | 0.51 |
| 5 | CLA | 0.02 |

**ICL Accuracy is predictable with progress measurements.** To assess sufficiency in addition to correlation, we train a random-forest regressor to predict ICL accuracy from the progress measurements. The results, shown in Tab. 2, demonstrate that the core ICL circuit is highly predictive in both settings. In the unimodal case, the two main components—$\text{PHStrength}_1^{(1)}$ and $\text{IndStrength}_2$—are sufficient to predict ICL accuracy with high fidelity ($R^2 = 0.909$). This predictive power holds in the multimodal setting, where the same core circuit remains the primary driver ($R^2 = 0.90$). In both stages, including $\text{TLA}_1$ (which measures general attention to label positions) further boosts performance to $R^2 = 0.96$, confirming its role as a complementary mechanism for locating relevant context. This shows that the same fundamental circuit underpins ICL in both unimodal formation and multimodal adaptation. Full analysis is deferred to App. A.3.3.

> **Takeaway.** The emergence of ICL is tightly coupled to a small set of interpretable attention patterns that form a quantifiable circuit. Unimodal pretraining focuses on *building* this circuit. Subsequent multimodal training focuses on *refining* this circuit.

## 4.5 DISCUSSION: MECHANISTIC INSIGHTS AND CONNECTIONS TO REAL MLLMS

**External validation of controlled findings.** To test whether our controlled results transfer to real MLLMs, we evaluate Qwen2.5-VL (3B/7B) (Team, 2025) and IDEFICS (9B/80B) (Laurençon et al., 2023) on the VL-ICL benchmark (Zong et al., 2024) and analyze attention using our progress measurements (Sec. 4.4) on Qwen2.5-VL. The 7B model consistently outperforms the 3B, supporting the view that scaling the decoder strengthens multimodal ICL (Sec. 4). Mechanistically, the induction heads are substantially stronger in 7B than in 3B, mirroring the positive correlation between the induction circuit strength and ICL accuracy. We provide detailed results in App. A.4, where stronger previous-token and induction heads coincide with stronger ICL.

**Connecting mechanisms to real MLLMs.** Our mechanistic findings help explain observed phenomena in production MLLMs: (1) Data asymmetry: The secondary modality's lower data require-

Table 2: A random forest regressor can predict final ICL accuracy ($R^2$) from our progress measurements. The core induction circuit is highly predictive in both settings, confirming its central role.

| Feature Subset | Unimodal $R^2$ | Multimodal $R^2$ |
|---|---|---|
| $\text{PHStrength}_1^{(1)}$, $\text{IndStrength}_2$ | $0.91 \pm 0.02$ | $0.90 \pm 0.01$ |
| $\text{PHStrength}_2^{(1)}$, $\text{TLA}_1$, $\text{IndStrength}_2$ | $0.96 \pm 0.02$ | $0.96 \pm 0.06$ |
| All metrics | $0.97 \pm 0.01$ | $0.98 \pm 0.01$ |

ments once the primary modality installs the ICL circuit explains why vision alignment needs far less data than text Liu et al. (2023): visual encoders need only map features into existing decoder space, not relearn in-context reasoning. (2) RoPE's induction tax: RoPE weakens induction patterns in low-data regimes, though sufficient data overcomes this. This suggests exploring hybrid encodings rather than abandoning RoPE's length-generalization benefits. (3) Encoder quality: The correlation between encoder validation accuracy and downstream ICL validates the practice of using strong pretrained visual backbones. (4) Circuit refinement: Multimodal training refines existing induction heads rather than forming new circuits. This explains why some MLLMs resort to majority-voting heuristics Baldassini et al. (2024): they inherit strong context-label prediction from language backbones but lack refined vision-to-label matching.

## 5 RELATED WORK

**Unimodal ICL.** First observed as an emergent LLM ability (Brown et al., 2020), ICL has since been studied in NLP and synthetic settings (Min et al., 2022; Akyürek et al., 2022; Wei et al., 2023; Von Oswald et al., 2023; Singh et al., 2023; Dong et al., 2024), with work probing the roles of training data and demonstrations (Chan et al., 2022; Shin et al., 2022; Yadlowsky et al., 2023; An et al., 2023; Liu et al., 2024). Mechanistic studies have focused on induction circuits, first discovered by Elhage et al. (2021) and further investigated across various settings (Olsson et al., 2022; Wang et al., 2023), alongside theoretical interpretations (Xie et al., 2021; Singh et al., 2024). Reddy (2024) explored ICL emergence through both induction circuits and data distributional properties using minimal attention-only transformers, which particularly motivated our work. While these mechanistic analyses typically use simplified architectures that diverge from modern LLMs in key components, we further investigate more realistic structures and extend the analysis to the multimodal setting.

**Multimodal ICL.** MLLMs pretrained on interleaved image–text corpora exhibit multimodal ICL (Alayrac et al., 2022; Abdin et al., 2024; Hui et al., 2024; Zong et al., 2024). Work on strengthening this behavior includes instruction finetuning (Zhao et al., 2023; Doveh et al., 2024; Yu et al., 2024), learned shift vectors that approximate demonstration effects (Jiang et al., 2025), vision-attention encouragement (Jia et al., 2025; Chen et al., 2025b), and task-vector composition (Huang et al., 2024). However, recent diagnostic work reveals that many apparent cases of multimodal ICL are largely *text-driven* (Chen et al., 2025a), and simply doing majority vote Baldassini et al. (2024), suggesting that despite advances in capability and evaluation, the fundamental mechanisms enabling true multimodal ICL remain poorly understood. We address this gap with controlled experiments to identify the circuits underlying multimodal ICL.

## 6 CONCLUSION

We provided a mechanism-level account of ICL in modern multimodal transformers using controlled synthetic tasks. In unimodal settings, we show that RoPE weakens induction circuits and scaling shifts capacity toward memorization. In multimodal settings, we uncovered asymmetries: a primary modality installs the core ICL circuit, while a secondary modality can reach comparable ICL with far less data complexity. Unlike the unimodal case, scaling the decoder consistently helps multimodal ICL. Progress measurements reveal that multimodal training principally refines the existing induction head for label matching rather than forming new circuits.

**Reproducibility statement.** We will release code, configuration files, and scripts to fully reproduce all experiments, including data generators (with fixed random seeds), training/eval pipelines, and log parsing for progress metrics (PHStrength, IndStrength, TLA/CLA). We provide exact hyperparameters, model variants (depth/heads, PE choice), optimizer settings, batch sizes, and training budgets, plus checkpoints and evaluation splits for IWL/ICL/label-swap. For multimodal experiments, we include links and version pins for the image encoders and instructions to regenerate Omniglot-style datasets and synthetic distributions.

**Ethics Statement.** Our study uses synthetic and publicly available, non-sensitive image/text data; we do not process personal data, and no identifiable human subjects are involved. We report model sizes to support awareness of environmental impact, and we provide ablations and smaller-scale settings to reduce replication cost.

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

# A APPENDIX

This appendix provides comprehensive methodological details, extended experimental results, and external validation of findings presented in the main text. Section A.1 defines the experimental setup (data generation, controls, and evaluation protocols) for both unimodal and multimodal settings; Section A.2 presents extended unimodal results including positional encoding comparisons, scaling analyses, and progress measurement correlations; Section A.3 reports multimodal experiments on real image data, encoder pretraining effects, cross-modal interaction analyses, and early-fusion training, confirming that induction heads drive multimodal ICL; and Section A.4 validates our findings on production MLLMs (Qwen2.5-VL and IDEFICS), demonstrating that decoder scaling enhances ICL performance and strengthens induction circuits in real-world models.

## A.1 PRELIMINARIES

### A.1.1 PRELIMINARIES IN THE UNIMODAL SETTING

In the unimodal setting, inputs consist of $N$ labeled tuples followed by a query (Fig. 10a). Figure 10b summarizes the synthetic data generation and distributional controls, and Fig. 10c details the IWL/ICL evaluation protocol. We use $N = 8$ unless for both unimodal and multimodal settings otherwise stated.

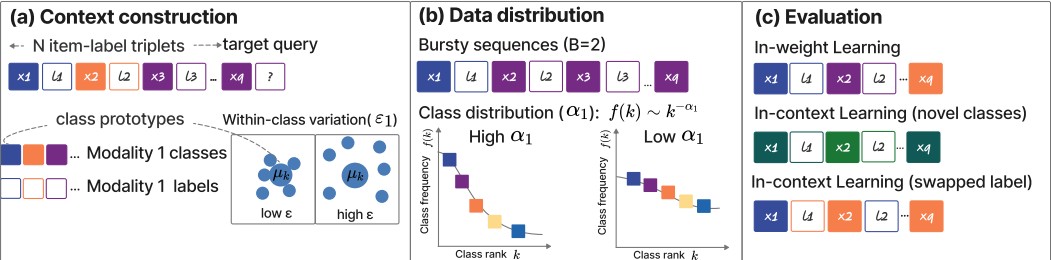

Figure 10: Overview of the preliminaries in the unimodal setting. **(a)** The context consists of $N$ item-label pairs $(x_i, l_i)$ followed by the target query. $K$ classes are assigned to $L$ labels. **(b)** The distributional properties for the synthetic data. Class instances are obtained by controlling within-class variation $\varepsilon$. Class frequencies follow a Zipfian distribution with exponent $\alpha$, and burstiness $B$ determines repeated class occurrences in context. **(c)** Evaluation distinguishes between in-weight learning, where test items belong to seen classes while not in the context, and in-context learning (ICL), where test classes are in the context but novel. A swapped-label condition further isolates ICL by permuting label assignments during evaluation.

### A.1.2 PRELIMINARIES IN THE MULTIMODAL SETTING

**Evaluation metrics.** We separate IWL from ICL to avoid conflating memorization with contextual reasoning, following Reddy (2024); Chan et al. (2022). Each test episode is a sequence of $N$ labeled exemplars (context) followed by a query; unless stated, we match training hyperparameters (e.g., $N$, burstiness $B$, Zipf exponent $\alpha$, within-class variation $\varepsilon$). **IWL:** sample both context items and the query i.i.d. from the *training* class distribution with the same priors; the query's class does not appear in the context examples. **ICL–Novel (primary test):** use novel classes unseen during training, provide labeled exemplars for those classes in the context, and draw the query from the same novel-class set; this measures the ability to infer class–label mappings from context alone. **ICL–Swap (label-swap control):** use training classes but apply a random permutation to context labels at test time and evaluate against the permuted mapping; this invalidates memorized mappings and isolates reliance on context. We report accuracy for IWL, ICL (average of ICL–Novel and ICL–Swap).

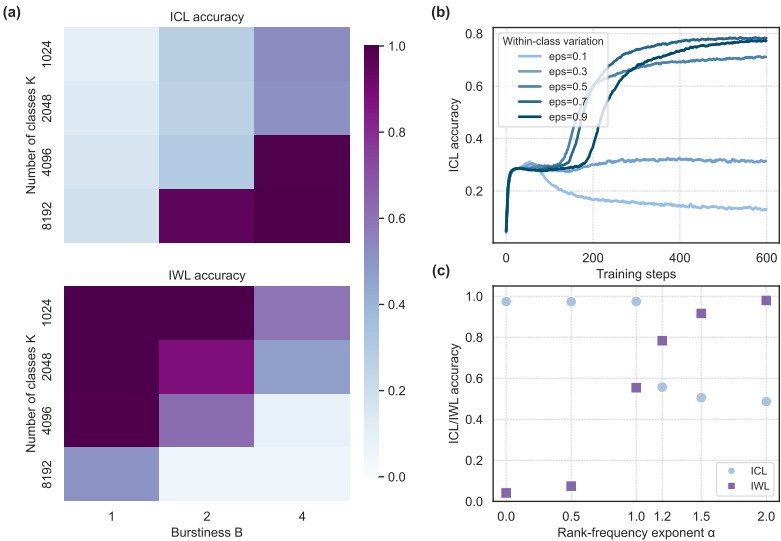

Figure 11: The data distributional findings transfer from the GMM setting. *(a) Number of classes and burstiness.* Larger number of classes $K$ and burstiness $B$ promotes ICL while decreasing IWL. *(b) Within-class variation.* Increasing $\varepsilon$ promotes ICL. However, it also slows down the emergence of ICL. *(c) Class-distribution skew.* Increasing $\alpha$ improves ICL. When $\alpha = 1$, a balance is achieved between ICL and IWL.

## A.2 EXTENDED RESULTS AND ANALYSIS IN UNIMODAL EXPERIMENTS

### A.2.1 UNIMODAL DATA DISTRIBUTIONAL PROPERTIES

As shown in Fig. 11 key trends are reproduced. In addition, our experiments reveal two extensions beyond prior work. First, strong ICL does not require burstiness in all the sequences: with $p_B < 1$, high $K$ and $B$ suffice, indicating that induction behavior learned from structured contexts transfers to unstructured ones. Second, while higher within-class noise $\varepsilon$ again promotes ICL, we additionally find that it slows convergence, as the model requires more data to form robust associations.

### A.2.2 IMPACT OF POSITIONAL ENCODINGS, CONTEXT LENGTH, AND DATA COMPLEXITY

To further investigate the mechanistic impact of positional encodings (PE) discussed in Sec. 3, we conducted a broader analysis comparing **Absolute PE (APE)**, **RoPE**, **ALiBi**, and a **Hybrid (APE + RoPE)** model. We swept two key variables: the context length (N, the number of item-label pairs) and the data complexity (proxied by $K \cdot \sqrt{B}$). The results are shown in Figure 12. First, for a fixed data complexity, ICL accuracy degrades for *all* PE types as the context length (N) increases. This suggests that longer contexts make it more difficult to find the matching exemplar, increasing the difficulty for the ICL circuit (as supported by the decreasing induction circuit strength in Figure 12). Secondly, We observe that ALiBi and RoPE perform similarly, and as argued in Sec. 3, both appear to weaken the formation of the offset-based induction circuits compared to APE. The Hybrid (APE + RoPE) model's performance lies in between as expected. Third, while APE and Hybrid PE demonstrate stronger ICL performance across most of the tested regimes, it is crucial to note the effect of data complexity. As seen in the high-complexity setting (Figure 12, right), the performance gap narrows significantly. At high complexity and short context lengths (e.g., N=8), it is possible for all four PE types to achieve near-perfect ICL accuracy.

These findings show that the design of RoPE and ALiBi provides a weaker inductive bias for the simple, offset-based induction circuits. This effect is most pronounced in low-data-complexity regimes, and *sufficiently high data complexity can compensate for this weaker bias. Importantly, they do not prevent ICL.*

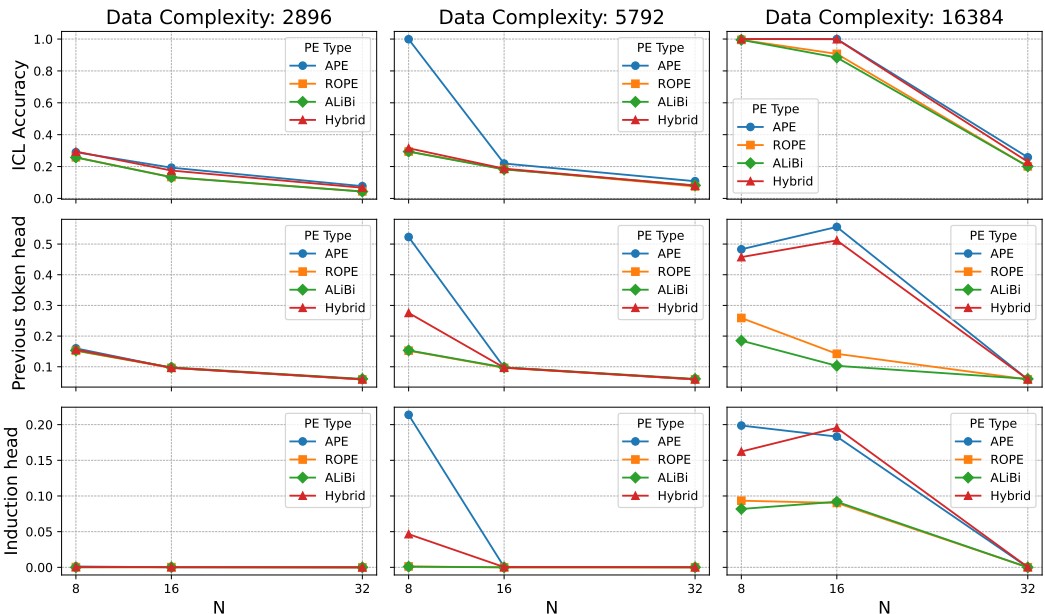

Figure 12: Comparison of the impact of different positional encodings across data complexity and context length on ICL accuracy and the strength of previous token head and induction head. Three key patterns are visible: **(1)** ICL accuracy degrades for all PE types as context length (N) increases. **(2)** ALiBi and RoPE cluster together, while APE shows the highest accuracy, with the Hybrid PE performing in between. **(3)** The performance gap between encodings narrows significantly in the high-complexity regime, where all PEs can achieve near-perfect accuracy at short context lengths.

### A.2.3 ICL AND IWL TRADEOFF IN MODEL SCALING.

The results in Fig.13 show that while both small and lager models can do ICL, scaling up both the number of heads and layers leads the model to favour in-weights memorization with a fixed data complexity.

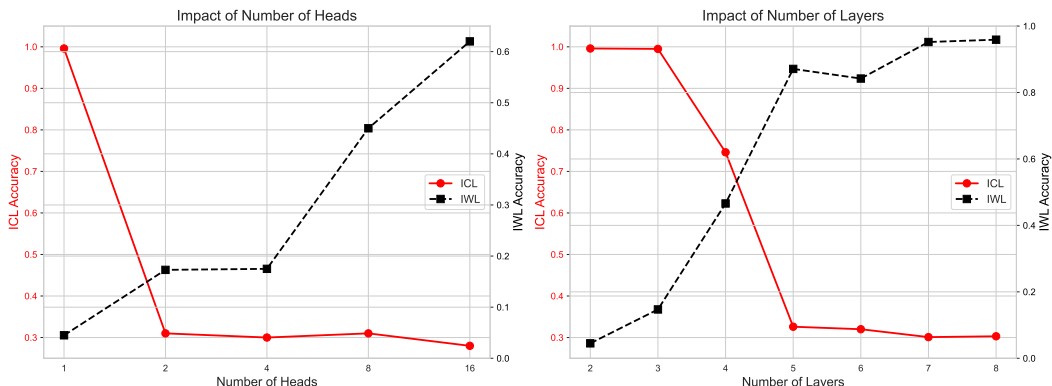

Figure 13: Scaling up both the number of heads and layers leads the model to favour in-weights memorization with fix data complexity ($K \cdot \sqrt{B} = 11585$).

| (a) Unimodal setting. | | |
|:---:|:---|:---:|
| **Rank** | **PM** | $\rho$ |
| 1 | $\text{PHStrength}_1$ | 0.72 |
| 2 | CLA | 0.65 |
| 3 | $\text{IndStrength}_2$ | 0.61 |
| 4 | $\text{TLA}_1$ | 0.59 |
| 5 | $\text{TLA}_2$ | 0.11 |
| 6 | $\text{IndStrength}_1$ | 0.06 |
| 7 | $\text{PHStrength}_2$ | -0.10 |

| (b) Multimodal setting. | | |
|:---:|:---|:---:|
| **Rank** | **Metric** | $\rho$ |
| 1 | $\text{IndStrength}_2$ | 0.70 |
| 2 | $\text{PHStrength}_1^{(1)}$ | 0.58 |
| 3 | $\text{TLA}_2$ | 0.56 |
| 4 | $\text{TLA}_1$ | 0.51 |
| 5 | $\text{PHStrength}_1^{(2)}$ | 0.48 |
| 6 | $\text{PHStrength}_2^{(2)}$ | 0.47 |
| 7 | $\text{IndStrength}_1$ | 0.10 |
| 8 | CLA | 0.02 |
| 9 | $\text{PHStrength}_1^{(1)}$ | -0.02 |

Table 3: Pearson correlations between progress measurements and ICL accuracy in unimodal (a) and multimodal (b) settings. We consider $\rho > 0.5$ strong and $\rho > 0.7$ very strong.

### A.2.4 PROGRESS MEASUREMENT ANALYSIS

**Pearson Correlation.** The results in the Table 3a align with the standard induction-circuit picture: $\text{PHStrength}_1$ correlates most with ICL accuracy (previous-token copying), $\text{IndStrength}_2$ ranks highly (label matching), and CLA is also strong, reflecting prediction from context labels. CLA ranks second, indicating that high ICL models tend to predict labels seen in the context. TLA in the first layer shows moderate correlation with ICL accuracy. This suggests that, in addition to the previous token function, the first attention head also contributes to the model learning to predict the labels existing in the context. This is further supported by a Pearson correlation of 0.6 between $\text{TLA}_1$ and CLA, indicating that attention to all context labels in early layers may be a key mechanism behind the model's ability to select labels from the context.

**Random forest prediction.** The results in the Table 4 show that the combination of $\text{PHStrength}_1$ and $\text{IndStrength}_2$, which correspond to the two key components of the induction head circuit, achieves an $R^2$ score of 0.909. This result indicates that these two mechanistic metrics alone capture the majority of the variance in ICL accuracy across models trained with varying data distributions and architectures. Adding $\text{TLA}_1$ as a third feature further improves performance to $R^2 = 0.960$. This suggests that the attention to all label positions provides complementary information to the mechanistic induction circuit, likely supporting the model's broader ability to locate context-relevant labels. This interpretation is reinforced by the earlier finding that $\text{TLA}_1$ correlates moderately with both ICL accuracy and CLA. Using the full set of progress measurements, the regressor achieves an $R^2$ of $0.974 \pm 0.014$, indicating that ICL performance is not only correlated with, but also highly predictable from, internal attention behaviors. These results suggest that attention patterns such as previous-token copying, label-focused attention, and context alignment are not incidental byproducts of learning, but form a reliable, quantifiable foundation for in-context reasoning.

Table 4: Random-forest performance ($R^2$) for predicting unimodal ICL accuracy from subsets of progress measurements.

| **Feature subset** | **$R^2$** (mean $\pm$ std) |
|:---|:---:|
| $\text{PHStrength}_1$, $\text{IndStrength}_2$ | $0.91 \pm 0.02$ |
| $\text{PHStrength}_1$, $\text{TLA}_1$, $\text{IndStrength}_2$ | $0.96 \pm 0.02$ |
| All metrics | $0.97 \pm 0.01$ |

### A.3 EXTENDED RESULTS IN MULTIMODAL EXPERIMENTS

### A.3.1 DATA DISTRIBUTIONAL FINDINGS TRANSFER TO REAL IMAGE DATA.

We extend our experiments on Omniglot dataset, which has 1.6k handwritten characters. We pre-train the decoder on GMM data and construct the multimodal input by varying the number of classes, the burstiness, within-class variance (adding noise) ,and the class distribution. The results in Fig. 14 show that findings from the synthetic setting transfer.

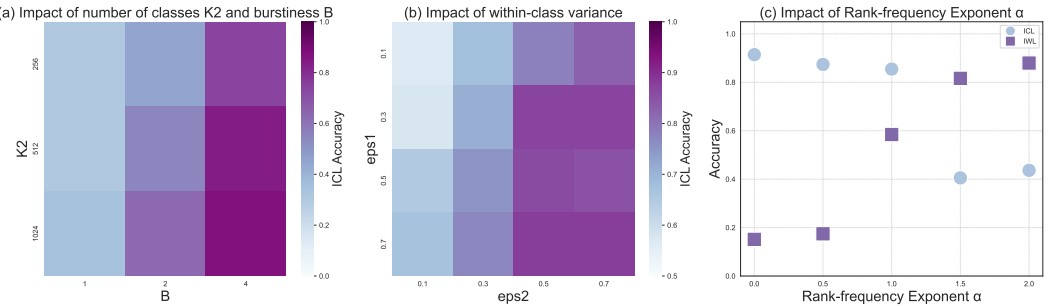

Figure 14: The data distributional findings transfer from the GMM setting to the **Omniglot** dataset. *(a)* Larger number of classes $K2$ and burstiness $B$ promotes ICL. The symmetry also transfers. When $K1 = 8192$, a smaller $K_2 = 256$ is enough for the model to learn a very good ICL. *(b)* Within-class variation.Increasing $\varepsilon_1$ $\varepsilon$and $\varepsilon_2$ promotes ICL. *(c)* Class-distribution skew. Increasing $\alpha$ improves ICL. When $\alpha_1 = \alpha_2 = 1$, a balance is achieved between ICL and IWL.

### A.3.2 HOW DOES THE PRETRAINING DATASET FOR THE ENCODER IMPACT DOWNSTREAM MULTIMODAL ICL?

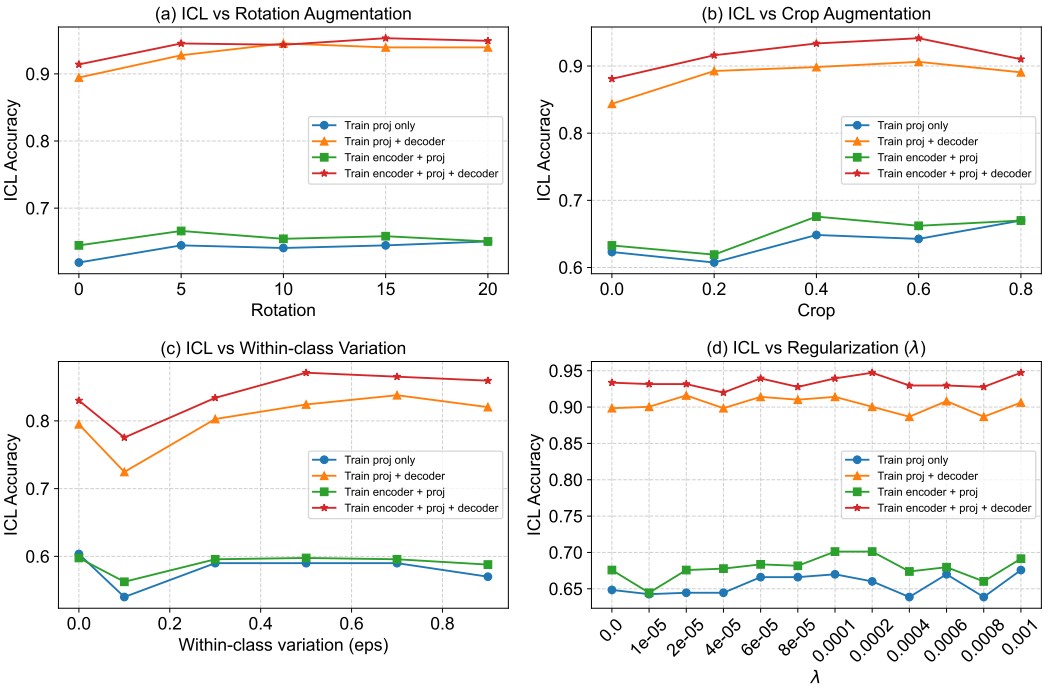

Figure 15: The correlation between statistical properties and regularization during encoder pretrain-ing with the downstream ICL does not show a clear pattern across different training regimes.

We pretrain a family of encoders on Omniglot with controlled difficulty (class noise, sample count, label skew, regularization), yielding a range of validation accuracies. Each encoder is attached to the same projector and pretrained decoder, then fine-tuned under identical regimes. Encoder

validation accuracy strongly predicts downstream multimodal ICL, whereas encoder size and the specific pretraining statistics do not show a clear independent effect (Figs. 15, 16).

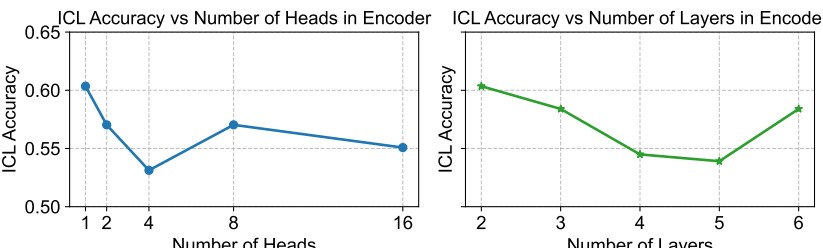

Figure 16: The encoder size shows no clear correlation with multimodal ICL accuracy.

### A.3.3 Progress measurement analysis

**Pearson Correlation.** As shown in the Table 3b, $\text{PHStrength}_1^{(1)}$ shows weaker correlation with ICL compared with $\text{IndStrength}_2$. During the unimodal pretraining stage, the model already learns a strong previous-token head in the first layer which is retained in the multimodal stage, as shown in the Fig. 9. It indicates that in the multimodal training stage, the dominant learning objective is to match the new modality to the correct class, making the continued development of the induction head the most sensitive indicator of progress. Similarly, $CLA$ exhibits a very low correlation with ICL accuracy because the model has learned in the first stage to predict the label from the context. In the multimodal training, $CLA$ maintains at a very high value as shown in the Fig. 9. $\text{TLA}_1$ continues to play a role as in the unimodal setting and interestingly, $\text{TLA}_2$ shows a moderately strong correlation with ICL accuracy ($r = 0.54$), but we find that it correlates even more strongly with $\text{IndStrength}_2$ ($r = 0.84$). This suggests that the predictive power of $\text{TLA}_2$ on ICL may be largely mediated through its alignment with $\text{IndStrength}_2$, which is the dominant indicator of ICL performance in this stage ($r = 0.68$). In this view, $\text{TLA}_2$ does not independently drive ICL, but rather co-develops with induction behavior. As the model learns to retrieve the correct label via contextual matching (induction), it also increases its attention to label positions more generally. This leads to a shared rise in both metrics, resulting in high intercorrelation and a secondary correlation with ICL. Thus, $\text{TLA}_2$'s relevance to ICL may be best understood as a supporting mechanism that reflects the formation of stronger induction behavior in the second layer.

**Random forest prediction.** The results in Table 5 confirm that the core induction mechanism remains the primary driver: using just $\text{PHStrength}_1^{(1)}$ and $\text{IndStrength}_2$ achieves a strong predictive performance with $R^2 = 0.9$. Adding $\text{TLA}_1$ improves the $R^2$ to 0.96. In contrast, adding $\text{TLA}_2$ results in a smaller gain ($R^2 = 0.92$), which supports our interpretation that its contribution is largely redundant with $\text{IndStrength}_2$. Using all available metrics yields $R^2 = 0.98$, showing that multimodal ICL accuracy can be predicted with high fidelity from a small set of interpretable attention behaviors.

Table 5: RF performance ($R^2$) using subsets of the progress measurements in the multimodal setting.

| Feature subset | $R^2$ (mean $\pm$ std) |
|---|---|
| $\text{PHStrength}_1^{(1)}$, $\text{IndStrength}_2$ | $0.90 \pm 0.01$ |
| $\text{PHStrength}_1^{(1)}$, $\text{IndStrength}_2$, $\text{TLA}_1$ | $0.96 \pm 0.06$ |
| $\text{PHStrength}_1^{(1)}$, $\text{IndStrength}_2$, $\text{TLA}_2$ | $0.92 \pm 0.01$ |
| All metrics | $0.98 \pm 0.01$ |

### A.3.4 CROSS-MODAL INTERACTION ANALYSIS VIA MODALITY ZEROING

To directly test whether the model performs genuine cross-modal reasoning or simply relies on a degenerate M1-only strategy, we conduct an ablation study where we selectively remove one modality at test time by replacing its tokens with zero vectors.

We take a model trained on multimodal sequences $x_1, x'_1, \ell_1, x_2, x'_2, \ell_2, \ldots, x_q, x'_q$ and evaluate it under three conditions:

- **Full (M1 + M2)**: standard evaluation with both modalities present
- **M1 only (M1 + zeros)**: replace all M2 tokens $x'_i$ with zero vectors
- **M2 only (zeros + M2)**: replace all M1 tokens $x_i$ with zero vectors

**Results.** Table 6 shows that zeroing either modality substantially degrades ICL accuracy, confirming that the model relies on both modalities during inference.

Table 6: Modality ablation results. ICL accuracy drops dramatically when either modality is removed, confirming genuine cross-modal integration.

| Input Type | M1 + M2 | M1 + zeros | zeros + M2 |
|---|---|---|---|
| ICL Accuracy | 0.967 | 0.336 | 0.063 |

Zeroing M2 reduces accuracy to 0.336 ($-65.2\%$), showing that removing the secondary modality's information substantially harms performance. Zeroing M1 is even more severe, reducing accuracy to 0.063 ($-93.5\%$), which indicates that while the induction circuit was initially installed by M1 during pretraining, M2 alone cannot recover ICL without the primary modality's contextual structure.

These results, combined with our progress measurement analysis in Sec. 4.4.2, demonstrate that multimodal ICL is not a degenerate M1-only process. The model genuinely integrates information from both modalities through the shared self-attention mechanism.

### A.3.5 EARLY-FUSION JOINT TRAINING

In the main paper, we train the decoder on M1 (unimodal pretraining) before introducing M2 via a projector (late fusion). To understand how the observed modality asymmetry depends on this training schedule, we conducted additional experiments with **early-fusion joint training**, where both modalities are trained together from scratch.

We initialize a transformer from scratch and train it jointly on both M1 and M2. Since M1 and M2 have different dimensionalities ($D_1 \neq D_2$), we use a simple projector to map M2 into M1's embedding space. The input sequence is $x_1, x'_1, \ell_1, x_2, x'_2, \ell_2, \ldots, x_q, x'_q$, where $x_i$ are M1 tokens and $x'_i$ are projected M2 tokens. We systematically vary the number of classes in each modality ($K_1, K_2$) and the burstiness $B$, then measure ICL and IWL accuracies.

**Results.** Tables 7–9 report ICL accuracy for $B \in \{1, 2, 4\}$, and Tables 10–12 report the corresponding IWL accuracies.

Table 7: ICL Accuracy with $B = 1$ (early-fusion joint training)

| $K_2 \backslash K_1$ | 2048 | 4096 | 8192 | 16384 |
|---|---|---|---|---|
| 2048 | 0.25 | 0.25 | 0.27 | 0.25 |
| 4096 | 0.28 | 0.26 | 0.25 | 0.26 |
| 8192 | 0.27 | 0.27 | 0.28 | 0.29 |
| 16384 | 0.26 | 0.27 | 0.29 | 0.31 |

The results reveal a reversal of the asymmetry observed in late-fusion training. In late fusion, where M1 pretraining installs the induction circuit before M2 is introduced, varying $K_1$ has a stronger effect on ICL than varying $K_2$. In early fusion, the opposite is true: varying $K_2$ shows a much stronger

Table 8: ICL Accuracy with $B = 2$ (early-fusion joint training)

| $K_2\backslash K_1$ | 2048 | 4096 | 8192 | 16384 |
|---|---|---|---|---|
| 2048 | 0.37 | 0.39 | 0.39 | 0.38 |
| 4096 | 0.37 | 0.39 | 0.37 | 0.35 |
| 8192 | 0.62 | 0.37 | 0.37 | 0.35 |
| 16384 | 0.52 | 0.65 | 0.50 | 0.57 |

Table 9: ICL Accuracy with $B = 4$ (early-fusion joint training)

| $K_2\backslash K_1$ | 2048 | 4096 | 8192 | 16384 |
|---|---|---|---|---|
| 2048 | 0.59 | 0.57 | 0.56 | 0.56 |
| 4096 | 0.57 | 0.58 | 0.59 | 0.56 |
| 8192 | 0.88 | 0.95 | 0.88 | 0.96 |
| 16384 | 0.89 | 0.87 | 0.58 | 0.88 |

effect on ICL performance across all burstiness levels. This reversal is explained by the positional structure of the sequence. In our early-fusion setup, each label $\ell_i$ is positioned immediately after the M2 token $x'_i$ rather than the M1 token $x_i$. Consequently, the simplest previous-token pattern the model can learn is to attend from $\ell_i$ to $x'_i$, and the simplest induction pattern for the query token is to locate the matching $x'_i$ (in M2) and move one step forward to retrieve the label. This architectural design naturally encourages the induction circuit to anchor on M2.

The primary-modality asymmetry observed in Sec. 4.1 is not a fixed property of the modalities themselves but emerges from the combination of pretraining schedule (which modality first learns the circuit) and sequence geometry (which tokens are positionally adjacent to labels). In late-fusion architectures with unimodal pretraining, the text modality becomes primary because it installs the induction circuit during language pretraining. Early-fusion joint training can shift this role to the vision modality if the sequence structure favors it.

### A.4 ADDITIONAL EVIDENCE ON PRODUCTION MLLMS

This appendix provides detailed results supporting Sec. 4.5. We evaluate Qwen2.5-VL (3B, 7B) (Team, 2025) and IDEFICS (9B, 80B) (Laurençon et al., 2023) on VL-ICL benchmark, a suite designed to directly measure the ICL ability of vision–language models Zong et al. (2024). We then compute the strength of the previous token head and induction head for Qwen2.5-VL (3B, 7B) to investigate whether the controlled findings on the correlation between ICL accuracy and the strength of the induction circuits hold in MLLMs. The results reinforce two conclusions from our controlled experiments: (i) scaling the decoder improves multimodal ICL, and (ii) the positive correlation between the ICL accuracy and the strength of the induction circuit transfers to real models.

**VL-ICL benchmark results.** Table 13 shows that Qwen2.5-VL-7B consistently outperforms the 3B variant across all six tasks. The consistent direction of change across diverse task types supports the scalability of multimodal ICL. The IDEFICS family exhibits an even clearer scaling signal: moving from 9B to IDEFICS2-80B raises average accuracy by +10.52 %. Together, these results indicate that larger decoders yield stronger in-context learning in multimodal scenarios, not just in language-only settings.

**Circuit analysis and head measurements.** We compute the strength of the previous token head and induction head for Qwen2.5-VL to identify the correlation between the induction circuit strength and ICL accuracy in the real MLLMs. For each attention head, we compute $\mathrm{PHStrength}(1)$ as the mean attention weight to the immediately preceding token and $\mathrm{IndStrength}$ as the attention that tokens place on positions right after any earlier occurrence of the same token—i.e., places where the model could copy from a matching in-context exemplar. Unlike Sec. 4.4, real MLLM inference doesn't have a single "target query" token, so we aggregate and average this induction-directed attention across all tokens in the sequence, rather than measuring it at one designated query position.

Table 10: IWL Accuracy with $B = 1$ (early-fusion joint training)

| $K_2 \backslash K_1$ | 2048 | 4096 | 8192 | 16384 |
|---|---|---|---|---|
| 2048 | 1.00 | 0.99 | 0.99 | 0.99 |
| 4096 | 0.98 | 0.97 | 0.97 | 0.98 |
| 8192 | 0.24 | 0.24 | 0.23 | 0.24 |
| 16384 | 0.10 | 0.10 | 0.10 | 0.09 |

Table 11: IWL Accuracy with $B = 2$ (early-fusion joint training)

| $K_2 \backslash K_1$ | 2048 | 4096 | 8192 | 16384 |
|---|---|---|---|---|
| 2048 | 0.97 | 0.98 | 0.98 | 0.97 |
| 4096 | 0.70 | 0.70 | 0.73 | 0.73 |
| 8192 | 0.09 | 0.12 | 0.15 | 0.13 |
| 16384 | 0.09 | 0.08 | 0.08 | 0.09 |

We evaluate these metrics on a batch of samples from the Open-MI task in the VL-ICL benchmark and rank heads by their average scores across samples. We show that scaling 3B→7B increases the ICL accuracy and also amplifies both components in the induction circuits: the strongest previous-token head increases from 0.105 to 0.127 (+20.8%), the strongest induction head from 0.009 to 0.011 (+25.3%); across the top-10, median gains are +2.3%(Prev) and +30.9% (Ind). These head-strength gains, especially IndStrength, track the VL-ICL accuracy improvements (Table 14).

> **Takeaway.** Scaling the decoder improves multimodal ICL in real MLLMs, and the positive correlation between the ICL accuracy and the strength of the induction circuit also transfers.

Table 12: IWL Accuracy with $B = 4$ (early-fusion joint training)

| $K_2 \backslash K_1$ | 2048 | 4096 | 8192 | 16384 |
|---|---|---|---|---|
| 2048 | 0.67 | 0.68 | 0.65 | 0.65 |
| 4096 | 0.46 | 0.41 | 0.38 | 0.43 |
| 8192 | 0.09 | 0.08 | 0.08 | 0.10 |
| 16384 | 0.07 | 0.07 | 0.09 | 0.08 |

Table 13: **VL-ICL performance on production MLLMs.** Accuracy (%) across six tasks comparing Qwen2.5-VL-3B/7B and IDEFICS-9B/IDEFICS2-80B. Both families improve with scale, supporting our finding that decoder scaling enhances multimodal ICL.

| Task | Qwen2.5-VL-3B | Qwen2.5-VL-7B | IDEFICS-9B | IDEFICS-80B |
|---|---|---|---|---|
| CLEVR | 5.50 | 7.50 | 30.33 | 32.43 |
| Matching-MI | 54.25 | 58.75 | 0.00 | 28.30 |
| Open-MI | 58.75 | 62.75 | 59.17 | 61.50 |
| Operator Induction | 8.35 | 8.50 | 14.44 | 21.67 |
| Operator Induction (Interleaved) | 10.00 | 11.35 | 15.00 | 36.67 |
| TextOCR | 46.50 | 48.50 | 28.00 | 29.50 |
| **Average** | **30.56** | **32.89 (+ 2.33 ↑)** | **24.49** | **35.01 (+ 10.52↑)** |

Table 14: **Top-10 head strengths by rank, side-by-side.** Each entry shows the layer–head index $(L, H)$ and strength $s$. "PHStrength$^{(1)}$" = previous-token behavior; "IndStrength" = induction behavior. The relative change is $\Delta\% = 100 \times \frac{s_{7B} - s_{3B}}{s_{3B}}$, so positive values mean the 7B head is stronger than the 3B head.

| | PHStrength$^{(1)}$ | | | IndStrength | | |
|---|---|---|---|---|---|---|
| Rank | Qwen-3B (L,H; s) | Qwen-7B (L,H; s) | $\Delta\%$ | Qwen-3B (L,H; s) | Qwen-7B (L,H; s) | $\Delta\%$ |
| 1 | (19,11; 0.1047) | (13,13; 0.1265) | **+20.8** | (17,10; 0.0091) | (15,20; 0.0114) | **+25.3** |
| 2 | (19, 1; 0.0738) | (15,11; 0.0626) | -15.2 | ( 5, 9; 0.0073) | ( 8,22; 0.0106) | +45.2 |
| 3 | (31, 9; 0.0623) | ( 1,15; 0.0570) | -8.5 | ( 5,14; 0.0060) | ( 4, 9; 0.0088) | +46.7 |
| 4 | ( 1, 1; 0.0582) | (10,22; 0.0530) | -8.9 | ( 3, 6; 0.0059) | ( 4,11; 0.0087) | +47.5 |
| 5 | (19, 5; 0.0516) | ( 0, 6; 0.0527) | +2.1 | (31,10; 0.0054) | (16, 9; 0.0086) | +59.3 |
| 6 | ( 2,13; 0.0491) | (15, 5; 0.0505) | +2.9 | (22,14; 0.0054) | (15,19; 0.0071) | +31.5 |
| 7 | (24,15; 0.0461) | (11,19; 0.0472) | +2.4 | ( 3,13; 0.0049) | ( 8,24; 0.0059) | +20.4 |
| 8 | (15, 9; 0.0456) | (21,15; 0.0470) | +3.1 | ( 6, 6; 0.0048) | ( 2,22; 0.0057) | +18.8 |
| 9 | (18,11; 0.0450) | ( 3,10; 0.0452) | +0.4 | (17,11; 0.0043) | ( 3,11; 0.0056) | +30.2 |
| 10 | (18,15; 0.0419) | ( 3,24; 0.0449) | +7.2 | ( 6, 9; 0.0043) | (27,13; 0.0050) | +16.3 |
| *Median* $\Delta\%$ | **+2.3↑** | | | **+30.9↑** | | |

