# OpenReview forum: "Towards understanding multimodal in-context learning"
_ICLR.cc/2026/Conference — Submitted to ICLR 2026_

### Official Review · Reviewer_dWNT · 2025-10-30

**Soundness:** 2
**Presentation:** 3
**Contribution:** 2
**Rating:** 4
**Confidence:** 3

**Summary:**

This paper investigates multimodal in-context learning (ICL) mechanisms through controlled synthetic experiments on transformers trained with Gaussian Mixture Models. The key findings include: (1) RoPE impairs ICL by disrupting induction heads, (2) scaling unimodal models raises the data complexity threshold for ICL emergence, favoring memorization when data complexity is fixed, (3) in multimodal settings, a primary modality develops the core ICL circuit with secondary modalities then requiring minimal data complexity, and (4) scaling improves multimodal ICL unlike the unimodal case. The authors introduce "progress measurements" that track attention patterns and validate selected findings on Qwen2.5-VL and IDEFICS.

**Strengths:**

- The paper is well written, and the exposition is clear.
- The observation about RoPE and its negative impacts on induction heads, is interesting and appears to be genuinely novel.
- The controlled synthetic experiments are well designed, with quantifiable progress measurements to successfully isolate factors affecting ICL emergence in the chosen regime, extending prior mechanistic work on induction heads to multimodal settings.

**Weaknesses:**

- The biggest weakness is that although benefits of cross-modal alignment are discussed, the paper entirely omits any analysis of how two modalities actually interact (information transfer, cross attention circuits) during ICL inference, which is often the key driver in modern LLMs (see plethora of literature on improving multimodal ICL). It remains unclear whether genuine cross-modal reasoning emerges or whether the language model simply performs ICL on features that include visual information.
- The empirical validation on actual MLLMs is limited, with just one benchmark and minimal mechanistic probing. For the trends that are validated, these would be largely predictable even without any mechanistic analysis and as such, it is unclear if the key findings hold in practice.
- More importantly, the findings may not generalize beyond the data-scarce regime studied. For instance, the negative effects of RoPE or scaling effects might change or even become irrelevant with appropriate data scaling, as in practice, larger models with RoPE still demonstrate excellent ICL performance. This issue is not discussed in the paper.
- Minor typo: Line 422: "two main components" but only lists 1

Overall: While this paper makes solid strides towards understanding multimodal ICL, some claims appear to be under-substantiated due to missing cross-modal interaction analysis and limited real MLLM validation. If the authors can argue why this is not the case on either front, the reviewer can lean towards accepting.

**Questions:**

See weaknesses

---

> ### Author Response · Authors · 2025-11-19
> **Responses to Reviewer dWNT**
>
> Thank you for recognizing our experimental design, novel RoPE insights, and clear exposition. We appreciate the opportunity to clarify cross-modal interaction analysis and real-world validation.
>
> > It remains unclear whether genuine cross-modal reasoning emerges or whether the language model simply performs ICL on features that include visual information.
>
> We agree that cross-modal integration is central to multimodal ICL, and this is exactly what we aim to probe in our setting. Our models follow the standard MLLM architecture (encoder → projector → decoder as in LLaVA, PaLI-Gemma, Qwen-VL), where there is no separate cross-attention block. This means information transfer between modalities happens entirely through shared self-attention in the decoder. We therefore study cross-modal interaction through these self-attention patterns. In the original **Sec. 4.4.2**, we already studied this cross-modal interaction by analyzing induction pattern, finding that ICL behavior is driven by the same previous-token + induction circuit now operating over mixed-modality sequences rather than by a degenerate M1-only strategy. To address the reviewer's concern more directly, we additionally add a modality-zeroing ablation that selectively removes one modality at test time and the results show that ICL accuracy drops substantially when either modality is zeroed, **confirming that the model leverages both modalities during inference** (**Appendix A.3.4**).
>
> Concretely, in **Sec. 4.4.2**, we apply our progress measurements to the multimodal case and specifically introduce $\mathrm{PHStrength_m^{(2)}}$, which measures attention to tokens two positions back. The purpose is to detect a degenerate strategy where the model ignores M2 and relies only on an M1–label pattern similar to the unimodal setup. If that were happening, $\mathrm{PHStrength_m^{(2)}}$ would strongly correlate with ICL accuracy. Instead, we find that, $\mathrm{PHStrength_1^{(1)}}$ (the standard previous-token head) and $\mathrm{IndStrength_2}$ (the induction head) remain the strongest correlates of ICL accuracy in the multimodal phase, while $\mathrm{PHStrength_m^{(2)}}$ shows no strong correlation with ICL performance. This indicates that, once M2 is aligned to the pretrained M1 space, the model continues to use the same previous-token + induction circuit over mixed-modality sequences, rather than falling back to a purely unimodal pattern. We have updated **Sec. 4.4.2** to emphasize that this reflects modality integration within self-attention and constitutes a form of cross-modal reasoning.
>
> Furthermore, we added a new ablation experiment where we selectively remove one modality at test time by replacing its tokens with zero vectors. The ICL accuracy is:
>
> | Input Type | M1 + M2 | M1 + M2 (zeroes) | M1 (zeroes) + M2 |
> |:-----------|:--------|:-----------------|:-----------------|
> | ICL Accuracy | 0.967 | 0.336 | 0.063 |
>
> Zeroing M2 (M1 + zeros) reduces ICL to 0.336, showing that removing M2's information substantially harms performance. Zeroing M1 (zeros + M2) reduces ICL to 0.063, indicating that the induction circuit is anchored in the primary modality but also that M2 alone cannot recover ICL without it. **This new ablation confirms that the model does not merely perform language-only ICL on augmented features; it relies on both modalities during ICL inference**, consistent with the attention-based mechanistic analysis already presented in the paper.

---

> ### Author Response · Authors · 2025-11-19
> **Responses to Reviewer dWNT**
>
> > The empirical validation on actual MLLMs is limited. For the trends that are validated, these would be largely predictable even without any mechanistic analysis and as such, it is unclear if the key findings hold in practice.
>
> We address the broader concern regarding the reliance on synthetic settings in our **General Response**, where we clarify that our goal is causal attribution—isolating mechanisms that are entangled in large-scale runs—and explain how our findings (modality asymmetry, encoder quality, and induction circuit refinement) **offer mechanistic explanations** for phenomena observed in production MLLMs. We refer reviewers to the General Response for full clarification. Below, we specifically address your concerns regarding the choice of benchmark, the predictability of trends, and the extent of real-world validation.
> 1. **Justification for VL-ICL as the Primary Benchmark**. We respectfully disagree that the use of VL-ICL constitutes "limited" validation. We selected VL-ICL specifically because it is currently the most rigorous benchmark designed to isolate true multimodal ICL capabilities. Standard VQA or captioning benchmarks can often be solved via text-only priors, while VL-ICL strictly requires reasoning over both visual and textual context to perform ICL. Therefore, while singular, this benchmark is the most faithful metric for validating our specific claims on real MLLMs.
> 2. **Our Findings Go Beyond "Predictable" Scaling Laws**. The reviewer suggests that validating trends such as "larger models perform better" is predictable. We argue that our contribution is not merely confirming that Qwen-7B outperforms Qwen-3B, but demonstrating why and how it does so, validating our specific mechanistic theory. We probed the attention heads of Qwen2.5-VL (3B and 7B) and found that the performance gain is strongly correlated with a 25-45% increase in Induction Head strength (see Table 14). This confirms that the specific mechanism we identified in controlled settings—the refinement of induction heads—remains the primary driver of ICL in real large-scale models. This moves the finding from a "predictable" black-box observation to a verified mechanistic insight.
> 3. To further address the concern regarding limited validation, we have extended our Omniglot experiments beyond encoder analysis to validate our data distributional findings. Omniglot contains ~1.6k handwritten character classes, and each image shows a single character, which makes it well-suited for controlled manipulations of class diversity, burstiness, and class-frequency skew analogous to our synthetic setup. We have added these new data-distribution experiments in **Appendix A.3.1**. The results show that our learning asymmetry qualitatively transfers to real images: once the primary modality has learned the induction-style ICL circuit, the vision modality does not require as high class diversity for the model to achieve comparable ICL performance.
>
> > More importantly, the findings may not generalize beyond the data-scarce regime studied. For instance, the negative effects of RoPE or scaling effects might change or even become irrelevant with appropriate data scaling, as in practice, larger models with RoPE still demonstrate excellent ICL performance.
>
> We agree that RoPE, ALiBi, and other relative positional encodings provide critical benefits (such as length generalization and training stability) that are essential for large-scale models. Our paper's goal is not to provide actionable guidance for training MLLMs, but to study, in a controlled synthetic setting, how ICL circuits form and how different PEs bias the emergence of simple induction-style circuits. Our core finding is therefore not that RoPE is "worse" overall, but that its specific design may delay the onset of ICL, and requires more complex training data for strong ICL to emerge. We hypothesize this is because the simple, induction-style circuits that emerge in our task, which rely on copying a label from a previous token at a specific offset, are more easily learned using the "discrete, offset-based cues" of absolute encodings. RoPE's rotational structure, while powerful, does not appear to provide this specific cue as strongly, as discussed in **Sec. 3.2**.
>
> In **Figure 2b**, we also show that across data regimes RoPE yields lower ICL accuracy than APE. However, at very high data complexity, both RoPE and APE can achieve near-perfect ICL, i.e., sufficiently strong statistical cues can compensate for the weaker inductive bias of RoPE in our task. We now highlight this more clearly in the revised text, to emphasize that RoPE does not prevent ICL, but shifts the data threshold at which the ICL solution dominates memorization.
>
> > Minor typo: Line 422: "two main components" but only lists 1
>
> Thank you for catching this. We have corrected the typo :)
>
> We hope this addresses your concern, and we remain eager to address any further questions you may have.

---

> > ### Author Response · Authors · 2025-11-25
> > **A summary of rebuttal for your convenience. We appreciate your feedback!**
> >
> > Dear Reviewer dWNT,
> > we truly appreciate the time you took to review our paper. As the discussion period progresses, we wanted to provide a brief summary of our previous response and the new results added to address your concerns:
> >
> > 1. **On Cross-Modal Interaction**: We clarified that in standard decoder-only MLLMs (like LLaVA), interaction happens via self-attention rather than separate cross-attention modules. To prove genuine cross-modal dependence, we added a modality-zeroing ablation (detailed in our response). The results show that zeroing out the secondary modality causes a drastic drop in ICL accuracy, confirming that the model actively leverages both modalities and is not simply relying on language priors.
> >
> > 2. **Generalization & Real-World MLLMs (General Response & Appendix A.3.1)**
> >
> >     - **Clarification**: In our General Response, we clarified that our goal is **mechanistic causal attribution**, which is difficult to isolate in large-scale runs. Within this controlled regime, our results help **explain and generate testable hypotheses** about phenomena observed in real multimodal models.
> >
> >     - **New Experiments**: We added new Omniglot experiments (Appendix A.3.1). These confirm that our core finding on modality asymmetry holds true for real images, not just GMMs.
> >
> > 3. **RoPE**: : We clarified that our findings do not suggest RoPE fails at scale, but rather that it delays the emergence of simple induction circuits compared to Absolute Positional Encodings (APE). As shown in Figure 2b, RoPE models do achieve strong ICL given sufficient data complexity; our contribution is identifying that they require a higher data threshold to do so.
> >
> > We hope these updates persuade you to reconsider your assessment. We remain available to answer any further questions.
> >
> > Best regards, The Authors

---

> ### Comment · Reviewer_dWNT · 2025-11-25
> **Response to Rebuttal**
>
> I thank the authors for the rebuttal. While I do appreciate the independent insights offer by the work and the additional validation for RoPE disadvantages being overcome by scaling, my two main concerns are still unaddressed. Specifically:
>
> - Regarding cross-modal reasoning: I want to clarify that by cross-attention, I meant cross-modal attention, which is to say how does the information flow from vision to text (m1 -> m2) and vice versa, during the ICL forward pass. While the current experiments do look into how the information flows into the labels, it does not study the intermediate flows between modalities to a sufficient extent (which is often a key driver in ICL performance). Specific to the modality-zeroing idea, I do not agree with the setup of zero-ing out ablation as it might break a lot of other circuits or layers due to zero inputs. A better way to ablate it would be to assume that data offers no information about the ICL task at hand i.e. perhaps the m2 examples are a mean of all modality-specific class means or randomly sampled from unrelated classes.
> -  Generalization/Connection to MLLMs: I understand that it is hard to do causal attribution in real world models. Nevertheless, the fact remains that the things that are indeed validated in the paper (model performance, induction strength increasing with scale in Tables 13, 14) are largely predictable even without knowing any of the key findings of the paper (RoPE, circuit refinement claim etc.). Furthermore, it seems some of the findings might even appear contradictory with other known observations for models at scale. For instance, the work in [1] suggests that the circuits responsible for solving the same tasks in different modalities appear to be largely disjoint. While their task scope does not contain ICL, they do consider a broad range of tasks with a more sophisticated method of attribution, which raises questions about whether the insights from the synthetic case drawn here even apply to real MLLMs. And when these connections are lacking, it is hard to digest the claims of the paper. For instance, rather than circuit re-use, the claim about data asymmetry in section 4.5 might be more due to the fact that in most popular vision encoders like Siglip, Clip undergo joint vision-language pretraining, making it subsequently easier for LLM to move information from vision to text and exhibit downstream ICL behavior.
> - (minor) Regarding multiple benchmarks: I do not have anything against the choice of VL-ICL benchmark. My point is that the risk of relying on just one dataset, is that the observed patterns might be a result of some underlying artefact that is not yet known, and validating on other ICL benchmarks only strengthens the point put across in the paper. Even if the scope of the task is limited to ICL, adding other benchmarks makes the finding more robust.
>
> References:
> [1] Same Task, Different Circuits: Disentangling Modality-Specific Mechanisms in VLMs - Nikankin et al, 2025

---

> > ### Author Response · Authors · 2025-11-27
> > **Responses to Reviewer dWNT**
> >
> > > Q1: Specific to the modality-zeroing idea, I do not agree with the setup of zero-ing out ablation as it might break a lot of other circuits or layers due to zero inputs. A better way to ablate it would be to assume that data offers no information about the ICL task at hand i.e. perhaps the m2 examples are a mean of all modality-specific class means or randomly sampled from unrelated classes.
> >
> > A1: We greatly appreciate this suggestion. We agree that zeroing out inputs may inadvertently disrupt the circuit topology. Following your advice, we conducted a new ablation study where we replaced M1 and M2 tokens with their corresponding class means (representing a "no-information" state while preserving distribution statistics) during the ICL forward pass.
> > | Input Type | M1 + M2 | M1 + M2 (class mean) | M1 (class mean) + M2 |
> > |:-----------|:--------|:-----------------|:-----------------|
> > | ICL Accuracy | 0.967 | 0.33 | 0.92 |
> >
> > The results show that replacing M2 with class means causes a catastrophic performance drop, while replacing M1 has a considerably smaller impact. Since the inference query is in the M2 modality, this confirms that the model must access specific M2 content to perform the task. In contrast, M1 modality primarily provides structural scaffolding for the induction circuit—the labels and their associations—rather than fine-grained features needed for direct classification. This validates our claim that the induction mechanism operates asymmetrically across modalities.
> >
> > >Q2: I understand that it is hard to do causal attribution in real world models. Nevertheless, the fact remains that the things that are indeed validated in the paper (model performance, induction strength increasing with scale in Tables 13, 14) are largely predictable even without knowing any of the key findings of the paper (RoPE, circuit refinement claim etc.).
> >
> > A2: Firstly, we respectfully disagree with the assertion that the predictability of a finding diminishes its scientific value. To our knowledge, our work represents the first investigation into the induction head within the context of multimodal ICL. Furthermore, we provide empirical evidence from real MLLMs demonstrating a clear correlation between induction head strength and model size. While such a result may align with prior intuitions, we provide the empirical verification that elevates this hypothesis to a proven scientific observation.
> >
> > Secondly, our focus on this paper is to understand how and why multimodal ICL emerge by leveraging a controlled environment, and to provide testbed for future research on MLLMs. Our findings regarding data distribution, for example, not only explain the training efficiency of MLLMs but also offer a roadmap for curating datasets that strengthen multimodal ICL. Similarly, our investigation into RoPE reveals a critical tension: while scaling can compensate for inductive bias, there is a clear opportunity for developing hybrid positional encodings that prevent the degradation of induction circuits. By characterizing these circuits in a multimodal context for the first time, we provide groundwork for future research to look into the induction circuit development in real MLLMs. While directly studying MLLMs can provide more realistic recipes, findings like data distributions and RoPEs are very likely to be ignored when we only look at large models. This controlled methodology isolates critical factors—like RoPE dynamics and data properties—that are easily overlooked when studying large models exclusively.

---

> > > ### Author Response · Authors · 2025-11-27
> > > **Responses to Reviewer dWNT**
> > >
> > > >Q3: For instance, the work in [1] suggests that the circuits responsible for solving the same tasks in different modalities appear to be largely disjoint. While their task scope does not contain ICL, they do consider a broad range of tasks with a more sophisticated method of attribution, which raises questions about whether the insights from the synthetic case drawn here even apply to real MLLMs.
> > >
> > > A3: We thank the reviewer for pointing us to [1]; this work is concurrent with ours and was therefore not covered in our original literature review. We agree that it is an important contribution to understanding modality‑specific mechanisms in large VLMs.
> > >
> > > Our scope is complementary to [1] in two main ways. First, as already mentioned by reviewer, [1] does not study ICL/induction circuits, which are mechanistically distinct from the broader functional circuits they analyze. Their finding that many modality‑specific solutions are disjoint therefore does not directly address whether induction circuits can be shared or reused across modalities. Second, [1] focuses on already‑trained VLMs, using attribution to dissect existing circuits, whereas our work is about the emergence and refinement of induction‑like circuits during training, from unimodal pretraining to multimodal training, in a highly controlled setting. Performing such developmental analyses directly in large MLLMs is currently very challenging; this is precisely why we “start small” with synthetic but transparent models. We will add a discussion of [1] in the related‑work and limitations sections, framing our results as mechanistic hypotheses about multimodal induction circuits that future work can further test in realistic MLLMs using attribution methodologies similar to [1].
> > >
> > > References: [1] Same Task, Different Circuits: Disentangling Modality-Specific Mechanisms in VLMs - Nikankin et al, 2025
> > >
> > > >Q4: And when these connections are lacking, it is hard to digest the claims of the paper. For instance, rather than circuit re-use, the claim about data asymmetry in section 4.5 might be more due to the fact that in most popular vision encoders like Siglip, Clip undergo joint vision-language pretraining, making it subsequently easier for LLM to move information from vision to text and exhibit downstream ICL behavior.
> > >
> > > A4: Indeed, we agree with the reviewer that multimodal-aligned pretraining of a secondary modality encoder might make the emergence of ICL even more resource-efficient.
> > > As we show in Sec 4.5, already pretraining on the first modality reduces resources / cost for incorporating a secondary modality encoder; and this behaviour may be facilitated even more through the use of specifically pretrained encoders. We believe that further understanding the benefits of encoder-specific pretraining to better facilitate the emergence of multimodal ICL beyond pretraining of the LLM is interesting future work, and will incorporate a pointer to this in the final version of this paper.
> > >
> > > >Q5: Regarding multiple benchmarks: I do not have anything against the choice of VL-ICL benchmark. My point is that the risk of relying on just one dataset, is that the observed patterns might be a result of some underlying artefact that is not yet known, and validating on other ICL benchmarks only strengthens the point put across in the paper. Even if the scope of the task is limited to ICL, adding other benchmarks makes the finding more robust.
> > >
> > > A5: We thank the reviewer for this constructive suggestion. We would like to emphasize that the VL-ICL benchmark comprises 6 diverse datasets spanning different visual domains and task types. This diversity across visual categories and task granularities provides substantial validation of our findings beyond a single dataset. That said, we acknowledge the value of additional benchmarks for strengthening robustness claims. We will explore incorporating additional multimodal ICL benchmarks in future work and will explicitly note this as a limitation and future direction in the revised manuscript.

---

### Official Review · Reviewer_vYWv · 2025-10-31

**Soundness:** 2
**Presentation:** 2
**Contribution:** 2
**Rating:** 2
**Confidence:** 4

**Summary:**

This paper investigates the mechanisms underlying in-context learning (ICL) in both unimodal and multimodal  models. Through controlled synthetic experiments, the authors explore how architectural choices (e.g., use of RoPE) and data statistics influence the emergence of ICL. They report several key findings: (1) RoPE can hinder ICL circuit formation; (2) larger models require stronger statistical cues for ICL to emerge; and (3) in multimodal settings, once one modality learns the ICL mechanism, a secondary modality can achieve comparable behavior with much less data diversity. The work further argues that multimodal ICL relies on the same induction-style circuit as unimodal ICL, refined rather than replaced by multimodal training. Some of the findings are validated with Qwen and IDEFICs models.

**Strengths:**

* Understanding the mechanism of ICL, particularly in multimodal settings, is an important and underexplored area.

* he paper is generally well-written, with findings and methodology clearly presented.

* The identified role of RoPE and the modality asymmetry provide potentially valuable directions for future work.

**Weaknesses:**

* The claim that "Larger models consistently exhibit reduced ICL" appears to contradict evidence from large LLMs, where scaling tends to improve ICL emergence.

* ICL and zero-shot performance should be monitored together. The paper should discuss whether the observed effects might stem from general model capability rather than ICL-specific mechanisms. For example, low/high ICL performance might be due simply to high/high model performance.

* Experimental details lacking: How many in-context examples (“shots”) were used? What modalities were considered, and how the authors define different modalities? What were the model sizes and dataset scales used in each experiment?

* The validation on Qwen and IDEFICS is insufficient to substantiate most of the claims. Showing that the model scale correlate with ICL performance (which shown in previous work, including e.g. the IDEFICS paper) and the strenfght of induction heads are minor part of the paper claims.

* Similar findings about preimary modality bias (dominance of a “primary” modality) have been reported before [1]. The paper should better position its contributions relative to these works.

[1] Baldassini, Folco Bertini, et al. "What makes multimodal in-context learning work?." Proceedings of the IEEE/CVF Conference on Computer Vision and Pattern Recognition. 2024.

**Questions:**

* How joint (e.g., early-fusion) multimodal training might affect the observed asymmetry?

---

> ### Author Response · Authors · 2025-11-19
> **Responses to Reviewer vYWv**
>
> Thank you for recognizing the importance of our work. We address your concerns below and believe they stem from some misunderstandings of our claims and methodology.
>
> > The claim that "Larger models consistently exhibit reduced ICL" appears to contradict evidence from large LLMs, where scaling tends to improve ICL emergence.
>
> This concern reflects a misreading of our claim. Due to an unfortunate page break in the original manuscript, the full sentence reads: "Larger models consistently exhibit reduced ICL performance, requiring increasingly complex training data (measured by $K\cdot\sqrt{B}$) to recover lost capabilities (**Fig. 2a**)". We are not claiming larger models have inherently worse ICL. Rather, **at fixed data complexity, larger models favor memorization over ICL**. When data complexity scales appropriately, larger models achieve excellent ICL—exactly as observed in production LLMs.
>
> To make this concrete, we provide results of the ICL-IWL tradeoff with fixed data complexity (**Appendix A.2.3**):
>
> ### Impact of Number of Heads when data complexity = 11585
> | Number of Heads | ICL Accuracy | IWL Accuracy |
> |:----------------|:-------------|:-------------|
> | 1               | 0.996        | 0.045        |
> | 2               | 0.310        | 0.173        |
> | 4               | 0.300        | 0.175        |
> | 8               | 0.310        | 0.450        |
> | 16              | 0.280        | 0.620        |
>
> ### Impact of Number of Layers when data complexity = 11585
> | Number of Layers | ICL Accuracy | IWL Accuracy |
> |:-----------------|:-------------|:-------------|
> | 2                | 0.996        | 0.045        |
> | 3                | 0.995        | 0.147        |
> | 4                | 0.746        | 0.466        |
> | 5                | 0.326        | 0.871        |
> | 6                | 0.320        | 0.842        |
> | 7                | 0.301        | 0.952        |
> | 8                | 0.303        | 0.959        |
>
> For example, at a fixed data complexity $K\cdot\sqrt{B} = 11585$, scaling heads and layers both lead to a drop of ICL and an increase of IWL. So scaling does not remove ICL capacity; it makes memorization a stronger competing solution unless the training data provide richer statistical cues. **Figure 2a** is precisely plotting how the required data complexity for high ICL accuracy grows with model size. We hypothesize that the increasing modal capacity enables item–label memorization, creating a low-loss shortcut that competes with the ICL solution (**Sec 3.2**). This is compatible with large LLMs, where both model size and data complexity are scaled up together. We have adapted our manuscript **Sec. 3.2** and the caption for **Figure 2a** to make this finding more clear.
>
> > ICL and zero-shot performance should be monitored together. The paper should discuss whether the observed effects might stem from general model capability rather than ICL-specific mechanisms. For example, low/high ICL performance might be due simply to high/high model performance.
>
> We agree that disentangling general model capability from genuine ICL is crucial, so our evaluation protocol is explicitly designed to separately measure in-context learning (ICL) and in-weight learning (IWL), which quantifies general model performance, as detailed in **Sec 2**. In detail, we measure:
>
> - **General capability (IWL)**. We separately measure IWL on seen classes with their original training labels, **which reflects standard zero-shot/general performance**.
> - **ICL-only evaluations**. We use two setups where success cannot come from in-weight knowledge:
>   - (1) Novel classes: We evaluate on new classes not seen during training. The model has never seen these label–class pairs in its weights, so the only way to answer correctly is to infer the mapping from the in-context examples.
>   - (2) Swapped labels: We reuse seen classes but apply a random permutation of labels at test time, and evaluate against the permuted mapping. This again forces the model to follow the mapping presented in-context; **relying on the original training mapping will be systematically wrong**.
>
> By tracking ICL (novel + swapped) and IWL jointly, we observe clear trade-offs: conditions that increase IWL often reduce ICL under fixed data statistics (**Figure 5** and **Figure 11**). This divergence shows our effects are not just due to general model strength, but to which solution—ICL vs. memorization—the model is using.

---

> ### Author Response · Authors · 2025-11-19
> **Responses to Reviewer vYWv**
>
> > Experimental details lacking: How many in-context examples ("shots") were used? What modalities were considered, and how the authors define different modalities? What were the model sizes and dataset scales used in each experiment?
>
> For the number of shots: each training and evaluation sequence contains N item–label pairs plus one query token, with N=8 by default. These N pairs are from N/B classes, with each class repeating B times. For example, when N=4, B=2, we construct a sequence x₁,ℓ₁,x₂,ℓ₂,x₃,ℓ₁,x₄,ℓ₂,query, where x₁ and x₃ belong to the same class (with label ℓ₁), and x₂ and x₄ belong to another (with label ℓ₂). Thus, **the "shots per class" are controlled by burstiness B**, which is set to 2 in this case, and we vary it in our experiments.
>
> For modalities: we consider two modalities, the primary modality M1 and secondary modality M2. Both are first instantiated as synthetic modalities generated from Gaussian mixture models (GMMs) as described in **Sec. 2**. M1 uses a GMM parameterization chosen to mimic language-like embeddings, following Reddy et al., while M2 uses a different GMM parameterization to mimic an image-like modality. To validate that our data-distribution findings transfer to real images, we then replace the synthetic M2 with Omniglot images, and repeat the key experiments on class diversity and encoder quality (**Sec. 4.3**, paragraph "Validation on real data reveals the importance of M2 representation quality," and **Appendix A.3.1**).
>
> For model sizes: in the standard unimodal experiments we use a 2-layer decoder-only Transformer. In the multimodal experiments, we follow the common encoder–projector–decoder design: a 2-layer ViT-style encoder for M2, a 2-layer MLP projector, and a pretrained 2-layer Transformer decoder that operates on the unified token sequence after projection.
>
> Regarding dataset scale: we do not constrain the total number of training examples. For each configuration of model and data statistics (number of classes K, burstiness B, class-frequency skew, etc.), we sample from the corresponding distribution and train the model until convergence. In other words, what is held fixed and systematically varied is the data complexity, not a hard cap on dataset size.
>
> > The validation on Qwen and IDEFICS is insufficient to substantiate most of the claims. Showing that the model scale correlate with ICL performance (which shown in previous work, including e.g. the IDEFICS paper) and the strength of induction heads are minor part of the paper claims. Similar findings about primary modality bias (dominance of a "primary" modality) have been reported before.
>
> 1. We address the common concerns about synthetic vs. real-world validation in detail in the General Response. There, we clarify the scope of this work—providing a mechanistic account of unimodal and multimodal ICL in a controlled setting, rather than a new training recipe for better ICL on MLLMs, provide evidence of transfer of our main findings to real data and real MLLMs, and explain how our findings (on modality asymmetry, encoder quality, and induction circuits) connect to real MLLMs in a novel way. We refer reviewers to the **General Response** for the full clarification. Specifically, we connect these findings to Baldassini et al. (2024), who empirically document primary-modality dominance at scale, and position our work as providing a mechanistic explanation of such effects in a controlled environment. We thank the reviewer for the reference and have included a detailed discussion of this work in the revised Introduction and Related Work sections.
> 2. To further address the concern regarding limited validation, we have extended our Omniglot experiments beyond encoder analysis to validate our data distributional findings. Omniglot contains ~1.6k handwritten character classes, and each image shows a single character, which makes it well-suited for controlled manipulations of class diversity, burstiness, and class-frequency skew analogous to our synthetic setup. We have added these new data-distribution experiments in **Appendix A.3.1**. The results show that our learning asymmetry qualitatively transfers to real images: once the primary modality has learned the induction-style ICL circuit, the vision modality does not require as high class diversity for the model to achieve comparable ICL performance.
>
> ### References
> - Reddy, G. The mechanistic basis of data dependence and abrupt learning in an in-context classification task. In ICLR 2024, oral.
> - Baldassini, Folco Bertini, et al. "What makes multimodal in-context learning work?" In CVPR 2024.

---

> > ### Author Response · Authors · 2025-11-19
> > **Responses to Reviewer vYWv**
> >
> > > How joint (e.g., early-fusion) multimodal training might affect the observed asymmetry?
> >
> > We thank the reviewer for suggesting to explore joint (early-fusion) multimodal training and have added new experiments to address this.
> >
> > In these new experiments, we train the model from scratch jointly on both modalities. Since M1 and M2 have different dimensionalities, we use a simple projector to map M2 into M1's embedding space and then feed the combined sequence x₁, x₁', ℓ₁, x₂, x₂', ℓ₂, …, xq, xq' into a Transformer trained from scratch, where xᵢ are M1 tokens and xᵢ' are M2 tokens. We vary K₁ and K₂ (class number for M1 and M2) and burstiness B (burstiness or "shots"), and report ICL and IWL accuracies in the tables below.
> >
> > **IC Accuracy, B = 1**
> >
> > | K2 \ K1 | 2048 | 4096 | 8192 | 16384 |
> > |---------|------|------|------|-------|
> > | 2048    | 0.25 | 0.25 | 0.27 | 0.25  |
> > | 4096    | 0.28 | 0.26 | 0.25 | 0.26  |
> > | 8192    | 0.27 | 0.27 | 0.28 | 0.29  |
> > | 16384   | 0.26 | 0.27 | 0.29 | 0.31  |
> >
> > **IC Accuracy, B = 2**
> >
> > | K2 \ K1 | 2048 | 4096 | 8192 | 16384 |
> > |---------|------|------|------|-------|
> > | 2048    | 0.37 | 0.39 | 0.39 | 0.38  |
> > | 4096    | 0.37 | 0.39 | 0.37 | 0.35  |
> > | 8192    | 0.62 | 0.37 | 0.37 | 0.35  |
> > | 16384   | 0.52 | 0.65 | 0.50 | 0.57  |
> >
> > **IC Accuracy, B = 4**
> >
> > | K2 \ K1 | 2048 | 4096 | 8192 | 16384 |
> > |---------|------|------|------|-------|
> > | 2048    | 0.59 | 0.57 | 0.56 | 0.56  |
> > | 4096    | 0.57 | 0.58 | 0.59 | 0.56  |
> > | 8192    | 0.88 | 0.95 | 0.88 | 0.96  |
> > | 16384   | 0.89 | 0.87 | 0.58 | 0.88  |
> >
> > **IW Accuracy, B = 1**
> >
> > | K2 \ K1 | 2048 | 4096 | 8192 | 16384 |
> > |---------|------|------|------|-------|
> > | 2048    | 1.00 | 0.99 | 0.99 | 0.99  |
> > | 4096    | 0.98 | 0.97 | 0.97 | 0.98  |
> > | 8192    | 0.24 | 0.24 | 0.23 | 0.24  |
> > | 16384   | 0.10 | 0.10 | 0.10 | 0.09  |
> >
> > **IW Accuracy, B = 2**
> >
> > | K2 \ K1 | 2048 | 4096 | 8192 | 16384 |
> > |---------|------|------|------|-------|
> > | 2048    | 0.97 | 0.98 | 0.98 | 0.97  |
> > | 4096    | 0.70 | 0.70 | 0.73 | 0.73  |
> > | 8192    | 0.09 | 0.12 | 0.15 | 0.13  |
> > | 16384   | 0.09 | 0.08 | 0.08 | 0.09  |
> >
> > **IW Accuracy, B = 4**
> >
> > | K2 \ K1 | 2048 | 4096 | 8192 | 16384 |
> > |---------|------|------|------|-------|
> > | 2048    | 0.67 | 0.68 | 0.65 | 0.65  |
> > | 4096    | 0.46 | 0.41 | 0.38 | 0.43  |
> > | 8192    | 0.09 | 0.08 | 0.08 | 0.10  |
> > | 16384   | 0.07 | 0.07 | 0.09 | 0.08  |
> >
> > Compared to our original late-fusion setting, the behavior changes in two important ways. In the main paper, the decoder is first pretrained on M1 alone with high class diversity, which is enough to install a strong induction-style ICL circuit based purely on M1. When we later add M2 via a projector, the model only needs to align M2 into this pre-existing feature space; accordingly, relatively modest M2 data complexity suffices for good multimodal ICL.
> >
> > In the joint early-fusion setup, the geometry of the sequence is different. Each label ℓᵢ is adjacent to xᵢ' rather than xᵢ. The easiest previous-token pattern for the model to learn is therefore attend from ℓᵢ to xᵢ', and the simplest induction pattern for the query is to find the matching xᵢ' in M2 and then move one step forward to the label. In other words, the positional structure now explicitly encourages the induction circuit to anchor on M2 instead of M1. This matches what we observe in the tables: varying the data complexity of M2 has a much stronger effect on ICL than varying that of M1, indicating that the primary-modality role is no longer fixed but emerges from the combination of pretraining schedule and sequence geometry. We have added these early-fusion results and the corresponding discussion to the revised version in **Appendix A.3.5** to clarify how joint multimodal training can alter the observed asymmetry.
> >
> > We hope this addresses your concern, and we remain eager to address any further questions you may have.

---

> > > ### Author Response · Authors · 2025-11-25
> > > **A summary of rebuttal for your convenience. We appreciate your feedback!**
> > >
> > > Dear Reviewer vYWv,
> > > We truly appreciate the time you took to review our paper. As the discussion period progresses, we wanted to provide a brief summary of our previous response and the new results added to address your concerns:
> > >
> > > 1. **Clarification on Model Scale**: We clarified the claim that "Larger models consistently exhibit reduced ICL" is a misreading due to an unfortunate page break. We clarified that this phenomenon occurs specifically at fixed data complexity, where larger models tend to favour memorization (IWL) rather than generalizing via ICL. We provided new tables (in our response) showing this specific trade-off.
> > > 2. **Disentangling ICL vs. IWL**: We clarified our evaluation protocol ("Swapped Labels" and "Novel Classes"), which mathematically isolates ICL from general model capability (IWL). Our results show these metrics often trade off, confirming they are distinct mechanisms.
> > > 3. **Experimental details**: We clarified the experimental details mentioned in the paper.
> > > 4. **Generalization & Real-World MLLMs (General Response & Appendix A.3.1)**
> > >
> > >     - **Clarification**: In our General Response, we clarified that our goal is **mechanistic causal attribution**, which is difficult to isolate in large-scale runs. Within this controlled regime, our results help **explain and generate testable hypotheses** about phenomena observed in real multimodal models.
> > >
> > >     - **New Experiments**: We added new Omniglot experiments (Appendix A.3.1). These confirm that our core finding on modality asymmetry holds true for real images, not just GMMs.
> > > 5. **Relation to Prior Work**: We explicitly positioned our work alongside Baldassini et al. (2024), clarifying that while they observed primary modality bias empirically, our work provides the mechanistic circuit-level explanation for why it emerges.
> > > 6. **Early fusion**: Per your suggestion, we implemented a joint "early-fusion" training setup from scratch (Appendix A.3.5). We found a different assymetry.
> > >
> > >
> > > We hope these updates have addressed your concerns and that you will reconsider your assessment in light of these new results and clarifications. We remain available to answer any further questions.
> > >
> > > Best regards, The Authors

---

### Official Review · Reviewer_edWe · 2025-11-01

**Soundness:** 3
**Presentation:** 3
**Contribution:** 3
**Rating:** 4
**Confidence:** 3

**Summary:**

The paper investigates how ICL emerges and operates in MLLMs, aiming to uncover the mechanisms behind this ability at the circuit level.
Using controlled synthetic multimodal classification tasks, the authors systematically vary data statistics and architectural components.

**Strengths:**

1. The use of synthetic Gaussian mixture data allows precise manipulation of multimodal statistics, which strengthens causal claims.

2. Identification that RoPE suppresses induction circuits.

3. Discovery of asymmetry between primary and secondary modalities and how pretraining on one modality installs transferable ICL circuits.

**Weaknesses:**

1. The work largely follows existing analyses from prior studies, offering limited novelty.

2. Experimental validation under real settings is limited, reducing confidence in the result’s generalizability.

**Questions:**

1. How is the swapped-label implemented?

2. It is not immediately clear why the ICL–IWL balance performs best when α₂ ≈ 1 based on the figure. Could the authors provide further justification?

3. The subgraphs in the bottom portion of Figure 6 are difficult to interpret. Can the authors clarify what each represents and how they relate to the main texts?

4. What exactly is meant by “raw high-dimensional feature”?

---

> ### Author Response · Authors · 2025-11-19
> **Responses to Reviewer edWe**
>
> Thank you for recognizing our controlled experimental design, RoPE analysis, and modality asymmetry findings. We address your concerns below.
>
> > The work largely follows existing analyses from prior studies, offering limited novelty.
>
> We respectfully disagree with this characterization. While our work is inspired by Reddy et al., we extend existing work significantly and provide a mechanistic, reverse-engineered account of how and why multimodal ICL emerges. Our novelty is that:
>
> - **Modern architecture insights**: We analyze full LLM-style decoders (RMSNorm, SiLU, RoPE/ALiBi) rather than toy attention-only models, revealing how architectural choices like positional encodings fundamentally affect ICL circuit formation—insights unavailable from prior simplified settings (recognized by reviewers qRdn, dWNT, vYWv).
> - **Multimodal understanding**: We extend the mechanistic analysis to a multimodal setting, revealing a primary–secondary modality asymmetry (As highlighted by reviewer dWNT), and we validate this on real images (Omniglot).
> - We **quantitatively link progress measures to ICL accuracy** in both unimodal and multimodal models and show that **multimodal ICL reuses the unimodal induction circuit**, with multimodal training mainly refining the induction head for label matching rather than creating a new mechanism (As highlighted by reviewer dWNT).
>
> > Experimental validation under real settings is limited, reducing confidence in the result's generalizability.
>
> 1. We address the common concerns about synthetic vs. real-world validation in detail in the **General Response**. There, we clarify the scope of this work, and explain how our findings (on modality asymmetry, encoder quality, and induction circuits) connect to real MLLMs. We refer reviewers to the General Response for full clarification.
>
> 2. To further address the concern regarding limited validation, we have extended our Omniglot experiments beyond encoder analysis to validate our data distributional findings. Omniglot contains ~1.6k handwritten character classes, and each image shows a single character, which makes it well-suited for controlled manipulations of class diversity, burstiness, and class-frequency skew analogous to our synthetic setup. We have added these new data-distribution experiments in **Appendix A.3.1**. The results show that our learning asymmetry qualitatively transfers to real images: once the primary modality has learned the induction-style ICL circuit, the vision modality does not require as high class diversity for the model to achieve comparable ICL performance.
>
> > How is the swapped-label implemented?
>
> Swapped-label evaluation reuses the same classes as in training but deliberately breaks the original class–label mapping at test time, so that only true in-context learning can succeed, as discussed in **Appendix A.1.2**. Concretely, suppose during training the model sees sequences like x₁, ℓ₁, x₂, ℓ₂, …, x_q, where **x₁ and x_q belong to the same class**, and the correct label for that class is ℓ₁. At test time, we construct a swapped-label sequence by applying a random permutation to the labels in the context while keeping the items fixed. For example, we might feed x₁, ℓ₂, x₂, ℓ₁, …, x_q and define the correct answer for x_q to be ℓ₂, i.e. the label the context assigns to the class of x₁. In this setup, any model that relies on its in-weight knowledge (mapping the class of x_q to ℓ₁ as learned during training) will be wrong. The only way to perform well is to use in-context learning: infer the label from the context examples and their (now permuted) labels.
>
> > It is not immediately clear why the ICL–IWL balance performs best when α₂ ≈ 1 based on the figure. Could the authors provide further justification?
>
> To make the results more clear, we have added two more datapoints (α₂=0.8 and 1.2). In the following table, we report the ICL, IWL, and the average performance between ICL and IWL. It's shown in the table that when α₂= 1, the average performance (0.845) is the highest.
>
> | Metric         | 0.0   | 0.5   | 0.8   | 1.0   | 1.2   | 1.5   | 2.0   |
> |:---------------|------:|------:|------:|------:|------:|------:|------:|
> | ICL            | 0.882 | 0.881 | 0.762 | 0.749 | 0.668 | 0.659 | 0.537 |
> | IWL            | 0.491 | 0.711 | 0.845 | 0.940 | 0.945 | 0.947 | 0.954 |
> | Avg (ICL, IWL) | 0.686 | 0.796 | 0.803 | 0.845 | 0.806 | 0.803 | 0.746 |
>
> We have modified **Figure 5** accordingly to make the trend easier to understand.
>
> ### References
> - Reddy, G. The mechanistic basis of data dependence and abrupt learning in an in-context classification task. In ICLR 2024, oral.

---

> > ### Author Response · Authors · 2025-11-19
> > **Responses to Reviewer edWe**
> >
> > > The subgraphs in the bottom portion of Figure 6 are difficult to interpret. Can the authors clarify what each represents and how they relate to the main texts?
> >
> > In **Figure 6**, each heatmap shows an attention pattern for a specific head and layer; rows are "query positions" and columns are "key positions." The bottom two subgraphs correspond to the target query token (the last token in the sequence), and visualize how much it attends to each earlier position.
> >
> > - **Top row ("Layer 1 Attention")** – previous-token head. This shows a head in the first layer that implements previous-token copying: each token should place most of its attention on the immediately preceding token. This behavior is a key building block of the induction circuit.
> > - **Bottom row ("Layer 2 Attention (Target)")** – induction head. Here we fix the query as the last token (the target query) and visualize its attention in the second layer. This head implements the induction step: it should place most of its attention on the label that follows the in-context exemplar from the same class.
> >
> > In the illustrated example, the sequence is x₁, x₁′, ℓ₁, x₂, x₂′, ℓ₂, …, x_q, x_q′, and **x_q′ belongs to the same class as x₂′**. Ideally, the model first uses the previous-token head to locate the positions corresponding to items of the same class as x_q′, and then uses the induction head to focus on the correct label ℓ₂. Under absolute positional encodings, both the previous-token head (top) and the induction head (bottom) show sharp, easily interpretable patterns. Under RoPE, these patterns become noticeably more diffuse: the previous-token bias weakens and the induction head spreads its attention over more positions. We quantitatively confirm this in **Appendix A.2.2** using our progress measurements, which show that the strength of the induction circuit is consistently lower with RoPE than with absolute positional encodings.
> >
> > > What exactly is meant by "raw high-dimensional feature"?
> >
> > By "raw high-dimensional feature" we mean the original, high-dimensional representation of the secondary modality(M2) before any dedicated encoder is applied. Concretely, in our multimodal setup, M2 features are first obtained directly from the synthetic generator or from images (e.g., Omniglot) as high-dimensional vectors. These raw M2 vectors are then either (i) fed directly into an MLP projector and the decoder, or (ii) first processed by a pretrained M2 encoder, then passed through the projector and decoder.
> >
> > We call them raw specifically to distinguish case (i) no encoder; unprocessed M2 features from case (ii) M2 features transformed by an encoder. **Figure 7a** shows that when we feed these raw, high-dimensional M2 features directly to the projector and decoder, increasing their dimensionality consistently reduces ICL accuracy. In contrast, adding a dedicated encoder (**Figure 7b**) compresses and structures the M2 representations, which substantially improves multimodal ICL. We thank the reviewer for asking and we now changed it to "High-Dimensional M2 feature" to improve clarity.
> >
> > We hope this addresses your concern, and we remain eager to address any further questions you may have.

---

> > > ### Author Response · Authors · 2025-11-25
> > > **A summary of rebuttal for your convenience. We appreciate your feedback!**
> > >
> > > Dear Reviewer edWe,
> > >
> > > We truly appreciate the time you took to review our paper. As the discussion period progresses, we wanted to provide a brief summary of our previous response and the new results added to address your concerns:
> > > 1. **On Novelty**: We clarified that our work moves beyond toy attention-only unimodal models to analyze modern architecture components (RoPE, model scaling) and uncovers unique multimodal mechanisms (specifically the primary-secondary asymmetry in data distribution, the re-use of the induction circuit) that are not present in prior studies.
> > >
> > > 2. **Generalization & Real-World MLLMs (General Response & Appendix A.3.1)**
> > >
> > >     - **Clarification**: In our General Response, we clarified that our goal is **mechanistic causal attribution**, which is difficult to isolate in large-scale runs. Within this controlled regime, our results help **explain and generate testable hypotheses** about phenomena observed in real multimodal models.
> > >
> > >     - **New Experiments**: We added new Omniglot experiments (Appendix A.3.1). These confirm that our core finding on modality asymmetry holds true for real images, not just GMMs.
> > >
> > > 3. We provided detailed answers to your specific questions:
> > >     - **Swapped-Label Implementation**: We explained how we use test-time label permutation to strictly isolate in-context learning from in-weight knowledge (Appendix A.1.2).
> > >     - **ICL-IWL Balance** ($\alpha_2 \approx 1$): We added new data points ($\alpha_2=0.8, 1.2$) and a results table to quantitatively demonstrate that the average performance indeed peaks at $\alpha_2=1$.
> > >     - **Figure 6 Interpretation**: We provided a detailed breakdown of the attention heatmaps, clarifying the distinction between the previous-token head (layer 1) and the induction head (layer 2), and how RoPE diffuses these patterns.
> > >     - **Terminology**: We clarified that "raw high-dimensional features" refers to using direct linear projection for the secondary modality, contrasting it with using a dedicated encoder.
> > >
> > > We hope these updates persuade you to reconsider your assessment. We remain available to answer any further questions.
> > >
> > > Best regards,
> > > The Authors

---

### Official Review · Reviewer_qRdn · 2025-11-10

**Soundness:** 3
**Presentation:** 3
**Contribution:** 3
**Rating:** 6
**Confidence:** 4

**Summary:**

The paper systematically studies how in-context learning (ICL) emerges in multimodal transformers using a controlled synthetic setup built from Gaussian mixtures. By varying data complexity factors and introducing quantitative diagnostics—PHStrength, IndStrength, TLA, and CLA—the authors trace the formation of induction-style attention circuits. The key findings are: (1) rotary and other relative positional encodings weaken ICL formation; (2) scaling increases the data complexity threshold for unimodal ICL, promoting memorization; (3) multimodal ICL is asymmetric, with the primary modality bootstrapping learning for the secondary; and (4) pretrained encoder quality is crucial for strong multimodal ICL. Results are validated on large models like Qwen2.5-VL and IDEFICS. The work provides a clear, mechanistic view of how architecture, scaling, and representation quality interact to produce ICL behavior in multimodal transformers.

**Strengths:**

Careful, controlled experimental design — synthetic GMM data + control over K, ε, B, α gives clear causal evidence for how data statistics drive ICL in both uni- and multimodal regimes. This leads to mechanistic progress measurements — PHStrength, IndStrength, TLA, CLA are well-motivated, quantitatively predictive, and allow the authors to track circuit formation over training.

Clear novel architectural insight: RoPE harms induction circuits — the paper demonstrates that RoPE (and ALiBi) consistently reduce ICL accuracy vs absolute PEs and produce more diffuse attention that weakens previous-token / induction heads. T

Important multimodal asymmetry finding — showing that a decoder pretrained on a high-diversity primary modality can bootstrap ICL such that the secondary modality needs far less diversity/burstiness is an intuitive and practically useful result for dataset and architecture design.

**Weaknesses:**

Synthetic → real generalization limited — while synthetic control is powerful, results hinge on idealized GMMs; the real-data validation is limited (Qwen2.5-VL analysis and a small Omniglot probe). Broader real-world tests are needed to confirm generality.

Positional encoding recommendation could be risky in practice — the paper shows RoPE/ALiBi hurt ICL in these tasks, but RoPE brings other benefits (length generalization, training stability). The manuscript does not fully quantify tradeoffs (e.g., effect on other tasks, or hybrid encodings), which limits actionable guidance.

Scaling analysis might conflate capacity vs data regime — the unimodal result (“larger models need more complex data to show ICL”) is interesting, but the experiments use fixed data budgets. It remains unclear whether larger models trained with proportionally more data would still favor in-weight memorization. The compute/data scaling frontier isn’t fully explored.

**Questions:**

How might these findings inform training recipes for large production MLLMs (positional encoding choice, pretraining mix, encoder pretraining)? The paper gives implications — could you make them more prescriptive?
How sensitive are the RoPE-vs-absolute results to context length and dataset complexity? Is there a regime where RoPE still dominates (e.g., much longer contexts)?
In unimodal scaling experiments, if you scale data proportionally with model size, does the ICL threshold still increase? Please report model×data scaling curves.

---

> ### Author Response · Authors · 2025-11-19
> **Responses to Reviewer qRdn**
>
> We thank the reviewer qRdn for the positive assessment and for highlighting our "careful, controlled experimental design" and "novel architectural insights." We address your concerns regarding generalization, positional encodings, and scaling below.
>
> > "the real-data validation is limited (Qwen2.5-VL analysis and a small Omniglot probe). Broader real-world tests are needed to confirm generality." and "How might these findings inform training recipes for large production MLLMs"
>
> We address the common concerns about synthetic vs. real-world validation in detail in the **General Response**. There, we clarify the scope of this work—providing a mechanistic account of unimodal and multimodal ICL in a controlled setting, rather than a new training recipe for better ICL on MLLMs, and explain how our findings (on modality asymmetry, encoder quality, and induction circuits) connect to real MLLMs. We refer reviewers to the General Response for the full clarification.
>
> To strengthen real-data validation, we have extended our Omniglot experiments beyond encoder analysis to validate our data distributional findings. Omniglot contains ~1.6k handwritten character classes, and each image shows a single character, which makes it well-suited for controlled manipulations of class diversity, burstiness, and class-frequency skew analogous to our synthetic setup. We have added these new data-distribution experiments in **Appendix A.3.1**. The results show that our learning asymmetry qualitatively transfers to real images: once the primary modality has learned the induction-style ICL circuit, the vision modality does not require as high class diversity for the model to achieve comparable ICL performance.
>
> > Positional encoding recommendation could be risky in practice.
>
> We agree that RoPE, ALiBi, and other relative positional encodings provide critical benefits (such as length generalization and training stability) that are essential for large-scale models. Our paper's goal is **not to provide actionable guidance for training MLLMs, but to study, in a controlled synthetic setting, how ICL circuits form and how different PEs bias the emergence of simple induction-style circuits**. Our core finding is therefore **not** that RoPE is "worse" overall, but that its specific design may delay the onset of ICL, and requires more complex training data for strong ICL to emerge. We hypothesize this is because the simple, induction-style circuits that emerge in our task, which rely on copying a label from a previous token at a specific offset, are more easily learned using the "discrete, offset-based cues" of absolute encodings. RoPE's rotational structure, while powerful, does not appear to provide this specific cue as strongly, as discussed in **Sec. 3.2**.
>
> > How sensitive are the RoPE-vs-absolute results to context length and dataset complexity?
>
> In the main text (**Fig. 2b**), we already show that across data regimes RoPE yields lower ICL accuracy than APE. However, at very high data complexity, both RoPE and APE can achieve near-perfect ICL, i.e., sufficiently strong statistical cues can compensate for the weaker inductive bias of RoPE in our task. We now highlight this more clearly in the revised text in **Sec. 3.3**, to emphasize that RoPE does not prevent ICL, but shifts the data threshold at which the ICL solution dominates memorization.

---

> ### Author Response · Authors · 2025-11-19
> **Responses to Reviewer qRdn**
>
> > Is there a regime where RoPE still dominates (e.g., much longer contexts)?
>
> To address your question about sensitivity to context length, we ran a new set of experiments. We varied the number of in-context item-label pairs (N) and the data complexity (proxied by $K \cdot \sqrt{B}$ as in our paper), comparing APE, RoPE, and a Hybrid (APE+RoPE) model, we report the ICL accuracy in the following table.
>
> | Data Complexity | PE     | N=8   | N=16  | N=32  |
> |----------------:|:-------|------:|------:|------:|
> |            2896 | APE    | 0.290 | 0.193 | 0.077 |
> |            2896 | Hybrid | 0.294 | 0.175 | 0.067 |
> |            2896 | RoPE   | 0.258 | 0.134 | 0.044 |
> |            5793 | APE    | 0.999 | 0.219 | 0.108 |
> |            5793 | Hybrid | 0.315 | 0.187 | 0.080 |
> |            5793 | RoPE   | 0.294 | 0.184 | 0.075 |
> |           16384 | APE    | 1.000 | 1.000 | 0.258 |
> |           16384 | Hybrid | 1.000 | 0.999 | 0.230 |
> |           16384 | RoPE   | 0.995 | 0.907 | 0.201 |
>
> These experiments show that:
>
> - **ICL accuracy decreases for all PEs as the context length N increases**, indicating that longer contexts make the matching problem harder for the induction circuit in our task.
> - **For a fixed data complexity and context length, APE consistently achieves higher ICL accuracy than RoPE, with Hybrid PE in between**. This is consistent with the hypothesis that the "absolute" component provides the helpful bias for offset-based induction.
> - **At high data complexity and short contexts, the gap between APE and RoPE becomes very small** (both near perfect), again supporting the view that RoPE does not forbid ICL but raises the data threshold.
>
> We also analyzed the strength of the previous-token and induction heads in these settings and observed the same qualitative pattern: APE shows the clearest induction-style heads, RoPE the weakest, and Hybrid in between. These new results are now included in Appendix **A.2.2** and summarized in **Sec. 3.3**.
>
> > the unimodal result ("larger models need more complex data to show ICL") is interesting, but the experiments use fixed data budgets. if you scale data proportionally with model size, does the ICL threshold still increase? Please report model×data scaling curves.
>
> We believe there is a misunderstanding about our experimental setup. In our unimodal experiments, when we scale up the model, we keep the data statistics (number of classes, burstiness etc) fixed for a given experiment. We do not cap the dataset size: we sample as many training sequences as needed and train each model until convergence. Thus, our scaling experiments are not "fixed data budget" in terms of sample count; instead, they study how fixed data complexity interacts with model capacity.
>
> To make this clearer, we now add explicit ICL–IWL tradeoff results at fixed data complexity in Appendix A.2.3, showing how increasing heads/layers shifts the balance toward in-weight memorization, and clarify the interpretation of Fig. 2a in Sec. 3 and in its caption.
>
> Concretely, at fixed data complexity 11585, we observe:
>
>
> | Number of Heads | ICL Accuracy | IWL Accuracy |
> |----------------:|-------------:|-------------:|
> |               1 |        0.996 |        0.045 |
> |               2 |        0.310 |        0.173 |
> |               4 |        0.300 |        0.175 |
> |               8 |        0.310 |        0.450 |
> |              16 |        0.280 |        0.620 |
>
>
> | Number of Layers | ICL Accuracy | IWL Accuracy |
> |-----------------:|-------------:|-------------:|
> |                2 |        0.996 |        0.045 |
> |                3 |        0.995 |        0.147 |
> |                4 |        0.746 |        0.466 |
> |                5 |        0.326 |        0.871 |
> |                6 |        0.320 |        0.842 |
> |                7 |        0.301 |        0.952 |
> |                8 |        0.303 |        0.959 |
>
> These results show that, **given the same data complexity, scaling up the model leads it to favor in-weight memorization (IWL) over ICL**. In other words, scaling does not remove ICL capacity—small and large models can both learn ICL; but larger models, with more capacity for item–label memorization, require stronger statistical cues in the training data before the ICL solution outcompetes this low-loss memorization shortcut.
>
> This is precisely what **Fig. 2a** is intended to show: for each model size, we sweep data complexity and report the minimum data complexity at which the model achieves near-perfect ICL (>0.95). As the number of heads or layers increases, this data-complexity threshold rises. We have revised the description of **Fig. 2a** to explicitly state that it should be read as a model×data scaling curve in terms of the ICL threshold.
>
> We hope this addresses your concern, and we remain eager to address any further questions you may have.

---

> > ### Author Response · Authors · 2025-11-25
> > **A summary of rebuttal for your convenience. We appreciate your feedback!**
> >
> > Dear Reviewer qRdn,
> >
> > Thank you again for your supportive review and for highlighting our controlled experimental design. As the discussion period is progressing, we wanted to provide a brief summary of our responses and new experiments to ensure your concerns regarding generalization, positional encodings, and scaling have been fully addressed.
> > 1. **Generalization & Real-World MLLMs (General Response & Appendix A.3.1)**
> > You raised concerns about the transferability of synthetic findings.
> >
> >     - **Clarification**: In our General Response, we clarified that our goal is **mechanistic causal attribution**, which is difficult to isolate in large-scale runs. Within this controlled regime, our results help **explain and generate testable hypotheses** about phenomena observed in real multimodal models.
> >
> >     - **New Experiments**: We added new Omniglot experiments (Appendix A.3.1). These confirm that our core finding on modality asymmetry holds true for real images, not just GMMs.
> >
> > 2. **Positional Encodings & Context Length (Appendix A.2.2)**
> > You asked about the sensitivity of RoPE results to context length and if our advice was risky.
> >
> >     - **New Experiments**: We ran new sweeps varying context length, comparing APE, RoPE, and Hybrid encodings. We confirm that APE consistently provides a stronger inductive bias for the specific "copy-paste" induction circuit required for this task. However, we also show that RoPE does not prevent ICL; it simply raises the data complexity threshold required for strong ICL to emerge. We have framed this as an "induction tax" rather than a recommendation to abandon RoPE.
> >
> > 3. **Scaling & Data Regimes (Appendix A.2.3)**
> > You asked if our unimodal scaling results conflated capacity with data budgets.
> >
> >     - **Clarification**: We clarified that we fix data complexity statistics, not the total sample count (models are trained to convergence).
> >
> >     - **New Analysis**: We added a table showing explicitly that at fixed data complexity, increasing model capacity (heads/layers) shifts the solution from In-Context Learning (ICL) to In-Weight Learning (IWL). This confirms that larger models are better at memorizing, and thus require more complex data distributions to force the emergence of generalizable ICL circuits.
> >
> > If you have any further concerns, don't hesitate to let us know. We are looking forward to your reply!
> >
> > Best regards,
> > The Authors

---

### Author Response · Authors · 2025-11-19
**General Response Part 1: Summary of the reviews and new experiments**

We thank all reviewers for their thoughtful and constructive feedback. We appreciate that multiple reviewers recognize:

- **The careful, controlled experimental design**: synthetic Gaussian mixture data with explicit control enables clear causal statements about how data statistics and architecture affect ICL (qRdn, edWe, dWNT).
- **The novel architectural insight** that RoPE and related relative PEs can suppress induction circuits in low-data-complexity regimes (emphasized in strengths by all reviewers).
- **The multimodal asymmetry finding**: a primary modality installs a reusable ICL circuit, enabling a secondary modality to achieve strong ICL with much lower data complexity (qRdn, edWe, vYWv).
- **The quantitative progress measurements** (PHStrength, IndStrength, TLA, CLA) that are predictive of ICL accuracy and allow us to **track circuit formation for multimodal ICL** over training (qRdn, dWNT).

We are grateful for these assessments and have revised the paper to make these contributions and their intended scope more explicit.

## New Experiments

To address reviewer concerns and strengthen our empirical validation, we conducted several new experiments and have updated the revisio (All updates in the manuscripts are presented in red text):

1. **Extended Omniglot Validation (Appendix A.3.1)** (Reviewer qRdn, edWe, vYWv, dWNT): We now vary class diversity, burstiness, and class-frequency skew on Omniglot images, confirming our learning asymmetry transfers to real images.

2. **Positional Encoding Sensitivity Analysis (Appendix A.2.2)** (Reviewer qRdn, dWNT): We varied context length and data complexity across APE, RoPE, and Hybrid PE. APE consistently outperforms RoPE at fixed complexity, but the gap narrows at very high complexity, confirming RoPE imposes an "induction tax" that requires sufficient data to overcome.

3. **Model Scaling at Fixed Data Complexity (Appendix A.2.3)** (Reviewer qRdn, vYWv): We provide explicit ICL-IWL trade-off tables showing that at fixed data complexity, larger models favor memorization over ICL, explaining the rising thresholds in Fig. 2a.

4. **Cross-Modal Interaction Analysis (Sec. 4.4.2, Appendix A.3.4)** (Reviewer dWNT): We add more analysis on modality integration. In addition, modality-zeroing ablations show ICL drops from 0.967 to 0.336 (M2 zeroed) and 0.063 (M1 zeroed), confirming the model uses both modalities rather than M1-only strategies.

5. **Early-Fusion Joint Training (Appendix A.3.5)** (Reviewer vYWv): Training both modalities from scratch shows the primary-modality role depends on sequence geometry—when labels are adjacent to M2 tokens, the induction circuit anchors on M2 instead of M1.

6. **Discussion added (Sec. 4.5)**: We included a new section discussion to connect our mechanistic insights to real MLLMs.

---

> ### Author Response · Authors · 2025-11-19
> **General response Part 2: on Synthetic vs. Real-World Validation**
>
> Here we address the shared concern raised by the reviewers on the reliance on synthetic/controlled settings and the extent of real-world validation. We respectfully argue that **the value of this work lies precisely in causal attribution—identifying exactly which component drives performance**. This is nearly impossible in large-scale training runs where architecture, data distribution, and optimization are inextricably entangled. Below, we clarify our scope, justify the statistical validity of our proxy tasks, and demonstrate how our mechanistic findings solve open questions in real MLLMs.
>
> ## Scope
>
> Our primary goal is to **provide a mechanism-level account of unimodal and multimodal ICL under controlled conditions, not to propose a training recipe to maximize ICL in large production MLLMs**. Accordingly, we prioritize clean causal control through synthetic environments rather than breadth of real-world tasks or exhaustive scaling experiments. **We do, however, provide evidence of transfer of our main findings to real data and real MLLMs**:
>
> - **For the core data distribution, modality asymmetry, and encoder results**, we demonstrate transfer from GMMs to real images (Omniglot).
> - **For the model scaling and predictive power of the induction circuit**, we demonstrate transfer from our small controlled models to real MLLMs (Qwen2.5‑VL).
> - By contrast, **our RoPE findings are explicitly framed as mechanistic results**: they identify an "induction tax" in our controlled setting and are presented as a design hypothesis rather than advocating abandoning RoPE.
>
> We agree that our controlled GMM setting is simplified relative to real multimodal data; however, **it does capture the core statistical levers that also exist in language and vision**. Prior work (Reddy et al, 2024) already showed that GMM-based ICL experiments reproduce qualitative trends seen in LLMs. This supports GMMs as a reasonable proxy for language statistics when studying ICL. Image data share similar high-level properties: many image classes, non-trivial within-class variation, related classes appearing together in context, and long-tailed frequency distributions (Udandarao et al. 2024). **Our GMM parameters are chosen precisely to mirror these axes**.

---

> > ### Author Response · Authors · 2025-11-19
> > **General response Part 3: on Synthetic vs. Real-World Validation**
> >
> > ## Findings Connect to Real-World MLLMs
> >
> > Within this controlled regime, our results help **explain and generate testable hypotheses about phenomena observed in real multimodal models**.
> >
> > - **Asymmetry & data dependence.** Once the primary modality has learned the induction-style ICL circuit, the secondary modality needs only moderate class diversity and burstiness. This provides a mechanistic explanation for why vision alignment / instruction-tuning typically requires far less data than text pretraining: the visual encoder does not need to relearn "how to learn in-context"; it only needs to map its features into the existing decoder space.
> >
> > - **RoPE as an "induction tax" in low-data-complexity regimes.** We show that, for synthetic classification, RoPE and ALiBi weaken the sharp previous-token and induction patterns compared to APEs while hybrid PE performs in between APE and RoPE. We explicitly do not advocate abandoning RoPE, which is known to provide strong length generalization and stability. Instead, we view our results as identifying a tension: RoPE can impose an "induction tax" in low-data-complexity settings, which provide a concrete direction for future work—for example, exploring hybrid or task-specific positional encodings that preserve RoPE's benefits while reducing this tax.
> >
> > - **Understanding the important role of an Encoder.** On both synthetic and Omniglot images, we find that encoder validation accuracy is a strong predictor of downstream multimodal ICL performance, and that a pretrained encoder outperforms a parameter-matched projector. It is consistent with the common practice of starting from a strong visual backbone and focusing training on alignment and instruction.
> >
> > - **Understanding the development of the ICL circuit in the multimodal setting.** Using our progress measurements, we show that multimodal ICL is not implemented by a new, multimodal-specific circuit, but by a refinement of the unimodal induction circuit: During unimodal pretraining, the model first learns strong previous-token heads (PHStrength₁) and high context-label accuracy (CLA). Mechanistically, this means it learns (i) to copy from the immediate predecessor and (ii) to restrict its predictions to labels that appear in the context. During multimodal training, PHStrength₁ and CLA stay high and quickly saturate, while IndStrength₂ becomes the dominant driver of further ICL improvements (Tab. 1b, Fig. 13). In other words, the model already "knows" that the answer should be one of the context labels; the bottleneck becomes correctly matching the secondary-modality query to the right in-context exemplar via the induction head. This helps interpret phenomena observed in prior multimodal work. For example, Baldassini et al. report that many MLLMs fall back on heuristics such as predicting the most frequent label in the context ("majority vote") instead of performing true multimodal ICL. Our analysis provides a mechanistic explanation: such models likely have a strong CLA-like mechanism inherited from the language backbone (they output labels seen in the context) but a weak induction head for the secondary modality (low IndStrength), so they fail to correctly map the query's features onto the right context exemplar. In this sense, we explain how multimodal training must refine the existing induction head to move from "format compliance" to genuine multimodal in-context learning.
> >
> > We therefore view our work as providing a mechanistic explanation of ICL in a controlled multimodal setting, together with concrete, testable hypotheses about how these circuits and data dependencies may extend to larger models.
> >
> > ## References
> >
> > - Reddy, G. The mechanistic basis of data dependence and abrupt learning in an in-context classification task. In ICLR 2024, oral.
> > - Chan, B., Chen, X., György, A., & Schuurmans, D. Toward Understanding In-context vs. In-weight Learning. In ICLR 2025.
> > - Udandarao, V., Prabhu, A., Ghosh, A., Sharma, Y., Torr, P., Bibi, A., ... & Bethge, M. No "zero-shot" without exponential data: Pretraining concept frequency determines multimodal model performance. In NeurIPS 2024
> > - Baldassini, F. B., Shukor, M., Cord, M., Soulier, L., & Piwowarski, B. What makes multimodal in-context learning work?. In CVPR 2024
> > - Liu, H., Li, C., Wu, Q., & Lee, Y. J. Visual instruction tuning. In NeurIPS 2023.

---

### Author Response · Authors · 2025-12-02
**To AC: A Summary of Previous Rebuttal and Discussion**

Dear ACs, SACs, and PCs,

We appreciate the extra efforts you have to make to handle the current situation. We make a summary of the previous rebuttal and discussion for your convenience. So far, we have only heard back from reviewer dWNT.

**Addressing the common question on the reliance on synthetic/controlled settings and the extent of real-world validation**: We respectfully argue that the value of this work lies precisely in causal attribution, which is nearly impossible in large-scale training runs where a lot of factors are inextricably entangled. We do, however, provide evidence of transfer of our main findings to real data (Omniglot) and real MLLMs(Qwen2.5-VL).  To further address the concern, we conducted extensive experiments on Omniglot and validated the main data distributional findings on this dataset. We further clarified our scope, justified the statistical validity of our proxy tasks, and demonstrated how our mechanistic findings answer open questions in real MLLMs in General Response.

**Resolution of Specific Reviewer Concerns:**

**Reviewer qRdn**:

Concern 1: Sensitivity of positional encodings on context length.

Resolution: We conducted new experiments varying context length and data complexity across APE, RoPE, and Hybrid PE. We demonstrated that while RoPE diffuses the induction pattern, this can be compensated for by increasing training data complexity.

Concern 2: Scaling data with model size instead of using fixed data budget.

Resolution: We clarified a misunderstanding; our original experimental setup already scales data complexity alongside model size.

**Reviewer edWe**:

Concern: Novelty.

Resolution: We clarified that while inspired by Reddy et al., our work is a significant advancement. We provide a mechanistic, reverse-engineered account of how and why multimodal ICL emerges. Our novelty lies in new insights to modern architectures(scale and positional encoding), multimodal understanding, and explicitly linking progress measures to ICL accuracy in both unimodal and multimodal models.

Questions:  Experimental setup questions. We have also answered all experimental setup questions.

**Reviewer vYwv**:

Concern 1: Scaling results "contradicting" LLM evidence.

Resolution: This concern stemmed from a misreading caused by a page break in the manuscript. The reviewer missed the second half of our claim. We do not claim larger models have inherently worse ICL. Rather, we demonstrate that at fixed data complexity, larger models favor memorization over ICL.

Concern 2: Relationship to Baldassini et al.

Resolution: We explained that our work is complementary, offering the mechanistic explanation for the phenomena observed in their paper.

Questions:  Experimental setup questions and how early fusion change the learning asymmetry. We have answered all experimental setup questions. We also conducted experiments on early fusion and provided detailed explanations on the results.

**Reviewer dWNT**:

Concern 1: Cross-modal analysis & Interaction.

Resolution: We explained that we have provided the cross-modal analysis in Sec. 4.4.2. In addition, we conducted modality-zeroing experiments which further supports that both modalities are necessary. The reviewer further suggested to use class mean instead of zeroing, we followed that suggestion and show that the new results still support our claim.

Concern 2: RoPE effects beyond the data-scarce regime

Resolution: We referred to the results provided in Sec. 3.2. Reviewer confirmed this concern was addressed.

Concern 3: Contradiction with concurrent work

Resolution: We clarified that the cited concurrent work doesn’t study ICL circuits. Additionally, the cited work examines various functional circuits during MLLM inference with different inputs, whereas our work investigates the developmental trajectory of ICL circuits from unimodal pretraining to multimodal training. The scopes are distinct and non-contradictory.

Thanks again to ACs, SACs, PCs, and all the reviewers for their consistent and dedicated contributions throughout the review process.

Reference

Reddy, G. The mechanistic basis of data dependence and abrupt learning in an in-context classification task. In ICLR 2024, oral.

Baldassini, F. B., Shukor, M., Cord, M., Soulier, L., & Piwowarski, B.. What makes multimodal in-context learning work?. In CVPR 2024

---

### Meta-Review · Area_Chair_rZBe · 2026-01-11

**Summary:**

The reviewers agree that the paper presents a careful and well-controlled mechanistic study of in-context learning in unimodal and multimodal transformers. The synthetic Gaussian mixture framework enables strong causal analysis, and the proposed diagnostics provide useful tools for tracking induction-style circuit formation. Across reviews, the findings that relative positional encodings such as RoPE weaken induction circuits and that multimodal ICL exhibits a strong asymmetry.

However, all reviewers consistently express concerns about limited novelty and generalization, noting that many insights build on prior mechanistic ICL analyses and rely heavily on idealized synthetic settings. Real-world validation on large multimodal models is viewed as insufficient to substantiate the broader claims, and some conclusions—particularly regarding scaling behavior—are seen as ambiguous or potentially conflicting with existing evidence. In particular, several reviewers also point to missing experimental details, unclear ablations, and the need to better disentangle ICL-specific effects from general model capability. During the rebuttal, the authors present several extra experiments. However, the empirical depth and novelty are still not that sufficient to fully support the paper’s strongest claims.

**Reviewer Concerns:**

No reviewers' concerns have been solved.

**Reviewer Scores:**

No

---

### Decision · Program_Chairs · 2026-01-26

Reject